# Tissue libraries enable rapid determination of conditions that preserve antibody labeling in cleared mouse and human tissue

**Theodore J Zwang[1,2,3]\*, Rachel E Bennett[1,2,3], Maria Lysandrou[1], Benjamin Woost[1], Anqi Zhang[4], Charles M Lieber[5], Douglas S Richardson[6], Bradley T Hyman[1,2,3]**

[1]Department of Neurology, Massachusetts General Hospital, Harvard Medical School, Boston, United States; [2]Harvard Medical School, Boston, United States; [3]Massachusetts Alzheimer's Disease Research Center, Charlestown, United States; [4]Department of Chemical Engineering, Stanford University, Stanford, United States; [5]Department of Chemistry and Chemical Biology, Harvard University, Cambridge, United States; [6]Department of Molecular and Cellular Biology and Harvard Center for Biological Imaging, Harvard University, Cambridge, United States

**Abstract** Difficulty achieving complete, specific, and homogenous staining is a major bottleneck preventing the widespread use of tissue clearing techniques to image large volumes of human tissue. In this manuscript, we describe a procedure to rapidly design immunostaining protocols for antibody labeling of cleared brain tissue. We prepared libraries of 0.5–1.0 mm thick tissue sections that are fixed, pre-treated, and cleared via similar, but different procedures to optimize staining conditions for a panel of antibodies. Results from a library of mouse tissue correlate well with results from a similarly prepared library of human brain tissue, suggesting mouse tissue is an adequate substitute for protocol optimization. These data show that procedural differences do not influence every antibody-antigen pair in the same way, and minor changes can have deleterious effects, therefore, optimization should be conducted for each target. The approach outlined here will help guide researchers to successfully label a variety of targets, thus removing a major hurdle to accessing the rich 3D information available in large, cleared human tissue volumes.

**\*For correspondence:** tzwang@mgh.harvard.edu

## Editor's evaluation

This valuable Tools and Resources paper presents a solid workflow for testing and comparing variations in tissue clearing, antigen retrieval, and antibody staining methods using thick slices of tissue. Though staining results vary sensitively with processing conditions, results from screening conditions in mouse brain tissue can be carried over to staining in human brain tissue. This solid story will be of broad interest to those carrying out immunohistochemistry experiments in human tissue samples.

## Introduction

In recent years, there have been substantial improvements in the techniques used to visualize features of brains, organs, and whole organisms in three dimensions. One promising method, tissue clearing, involves lipid removal and refractive index matching to significantly decrease light scattering. Tissue clearing allows for light microscopy to observe the distribution of individual molecules throughout entire organs millimeters to centimeters in size (*Ueda et al., 2020b*; *Richardson et al., 2021*). This

holds extraordinary promise for understanding human diseases, where the identification and characterization of abnormal structures is often an early step toward the development of clinical treatments (*Kim et al., 2016*). Difficulties with antibody staining and labeling is common in traditional thin tissue section histology, but is dramatically amplified in large, cleared samples. Although techniques have been developed to improve the homogeneity of staining (SWITCH; *Murray et al., 2015*) and increase the speed of antibody labeling in large, cleared tissue (eFlash [*Yun et al., 2019*], EFIC [*Na et al., 2021*], MDH [*Dwyer et al., 2021*]), these methods only work if an antibody's epitope is well preserved during the tissue preparation and clearing process. This is rarely the case and complete, specific and homogenous staining is now the most common bottleneck preventing the widespread use of tissue clearing in samples that cannot be modified to express fluorescent proteins such as human tissue. Therefore, most tissue clearing methods have been optimized for studies in rodents, and only a few recent studies apply clearing techniques to human tissue (*Scardigli et al., 2021*; *Lai et al., 2018*; *Susaki et al., 2020*; *Matos et al., 2010*; *Zhao et al., 2020*). Here, we present a simple optimization strategy and automated analysis routine that allows researchers to tune their immunohistochemistry (IHC) protocols for labeling protein targets in cleared and uncleared brain tissue. Further, we demonstrate that our optimized protocols can be extended from mouse to human tissue.

IHC has a history of being difficult, with myriad improvements taking place over decades to allow for its current ubiquity in pathology laboratories and biomarker discovery (*Matos et al., 2010*). The basic concept of IHC is to use antibodies to label specific antigens in a sample, which can then be imaged to determine the antigen's distribution. Successful IHC requires four fundamentals: (1) the antibody cannot denature, (2) the epitope cannot denature, (3) the antibody must diffuse completely through the tissue, and (4) the antibody must only bind its intended target. Often researchers develop lengthy protocols that include steps thought to enable successful labeling but are not necessarily based on empirical evidence or are borrowed from others using completely different antibodies or tissues. In general, IHC protocols share several common steps: first components of the tissue must be fixed in place to preserve its structure. This is typically performed by freezing the tissue or placing it in an aldehyde containing solution like formalin. It may then be stored for long periods of time before an investigation is initiated. Next, different methods of antigen retrieval can be deployed to enhance antibody binding. The tissue may also be blocked by incubating with serum or other agents to deactivate endogenous enzymes and reduce non-specific antibody staining. Finally, when labeling and imaging thick tissue, tissue clearing protocols are required that introduce several additional chemical treatments including dehydration, hydrogel embedding, delipidation, and immersion in a solution that is refractive index matched to the processed tissue (*Richardson et al., 2021*; *Richardson and Lichtman, 2015*).

It is well understood that differences in staining quality can result from biological heterogeneity such as small differences in tissue composition, individual antibodies, their epitopes, and overall tissue structure. However, further variability can be induced by each step of an IHC protocol (*Kim et al., 2016*; *Matos et al., 2010*; *Shi et al., 2011*). Because of this, IHC protocols can usually be reproduced with the same tissues, antibodies, and fluorophores. Unfortunately, it is common to have difficulties extending these protocols to different antibodies or tissues. The complexity of these multi-step protocols makes it prohibitively costly and time-intensive for researchers to optimize all possible IHC conditions for their specific study. This has slowed the testing and adoption of reported improvements in IHC procedures as many researchers are forced to guess at the best protocol for their samples.

Experiments with human tissue samples are essential for a complete understanding of the pathology of many diseases. However, the necessary optimization of IHC protocols described above is often not possible in human tissue due to its scarcity. Attempting to use IHC protocols optimized for different tissues or antibodies could confound results and limit the insight gained from such precious resources. Ideally, researchers could optimize protocols in a more plentiful model species such as mice and directly transfer these methods to human samples. However, it is not often clear whether IHC improvements in mice (*Murray et al., 2015*; *Wassie et al., 2019*; *Park et al., 2018*) would translate to better staining of human tissue. Methods that can rapidly determine optimal IHC protocols for individual antibodies and assess their applicability to tissues from different species are needed.

Inspired by our previous experience troubleshooting IHC in large volumes of mouse brain tissue, we developed a practical workflow to rapidly optimize IHC protocols for different antibodies. We then applied this methodology to a variety of targets in mouse and human brain tissue to understand (1)

how much variation exists if IHC protocols are optimized for individual antibodies and (2) if optimized IHC protocols can be generalized between species. We compare the effect of different IHC protocols on antibodies and a small molecule DNA label, 4',6-diamidino-2-phenylindole (DAPI). We also test the influence of protocols on different antibodies raised against the same protein target and finally, explore whether IHC protocols optimized in mouse tissue can produce equally high-quality staining in human tissue. Together this work provides a method to empirically determine the optimal procedure for IHC studies in large-volume tissue, assess its generalizability, and support the use of abundant mouse tissue to optimize IHC protocols for scarce human samples.

## Results

### Strategy to rapidly assess the optimization of protocols

Our workflow for determining an optimal IHC protocol is inspired by the success of quantitative structure-activity relationship (QSAR) modeling for drug discovery. QSAR models are used to predict the biological activities of untested drugs by first testing a library of drugs with similar but varied structures, measuring how each drug affects the activity, then creating a model that relates the structural changes to the observed activities (*Muratov et al., 2020*). Borrowing from this strategy and making a library of tissues that all went through similar processing steps but for a few key changes allows for us to predict the optimal staining protocol without needing to test every possible combination of conditions.

The use of large intact organs such as whole mouse brains is unnecessary and impractical for optimizing an IHC protocol. However, if the tissue sections are too thin, the impact of antibody diffusion cannot be fully appreciated and compared across protocols; whereas, if the sections are too thick, extensive incubation times are required. Through empirical testing, we found tissue sections 0.5–1.0 mm thick and antibody incubation times of 18–24 hr to be sufficient to assess the influence of our chosen conditions on IHC staining quality (*Figure 1*, *Figure 2*, *Figure 3—figure supplement 1*), and allows dozens of protocols or multiple antibodies to be tested with tissue from a single mouse brain depending on the epitope distribution. It should be noted that these data suggest the signal when staining 24 hr and under may be limited by the depletion of antibodies on their way through the tissue, which should be considered when making comparisons of staining quality.

Using QSAR modeling as a guide, we selected 17 conditions to test in our proof-of-principle studies (*Figure 1*). These conditions were divided across four standard IHC/tissue clearing steps: fixation, blocking/unmasking, delipidation, and antibody incubation. Five conditions were later omitted, and multiple 0.5–1.0 mm thick tissue sections were assigned to 12 protocols to create the final tissue library (*Figure 1*). With the library complete, a tissue sections from each condition could be selected for further head-to-head analysis in our image analysis pipeline that outputs a quantifiable metric of quality (see below). Using these metrics, we were able to predict the optimal combination of IHC conditions for each antibody we tested from initial fixation through to final labeling. During this proof-of-principle study, we only tested high-quality antibodies that were already known to have high specificity and limited off-target binding for IHC in thin tissue slices.

### Measuring IHC quality in cleared tissue

To compare the quality of IHC in the cleared tissue libraries we developed a data analysis pipeline to quickly extract the distribution of fluorescence in multiple samples and derive performance parameters. First, we used one of two different software (Imaris or Ilastik) to generate image segmentation masks. For analysis with Imaris, we isolated objects based on the intensity of fluorescence relative to nearby background (*Figure 2*). For analysis with Ilastik, we leveraged machine learning algorithms that allow users to classify pixels and supervise the generation of the segmentation mask (*Berg et al., 2019*). Either of these segmentation masks are then run through code in Matlab that extracts the signal, noise, signal-noise ratio, area of staining, and the distance into the tissue at which the staining intensity drops by half. IHC quality can then be made by comparing these features across samples. It should be noted that simply relying on depth to half-max staining does not separate the issues of free antibody penetration speed through the tissue nor the extent of antibody depletion along the way,

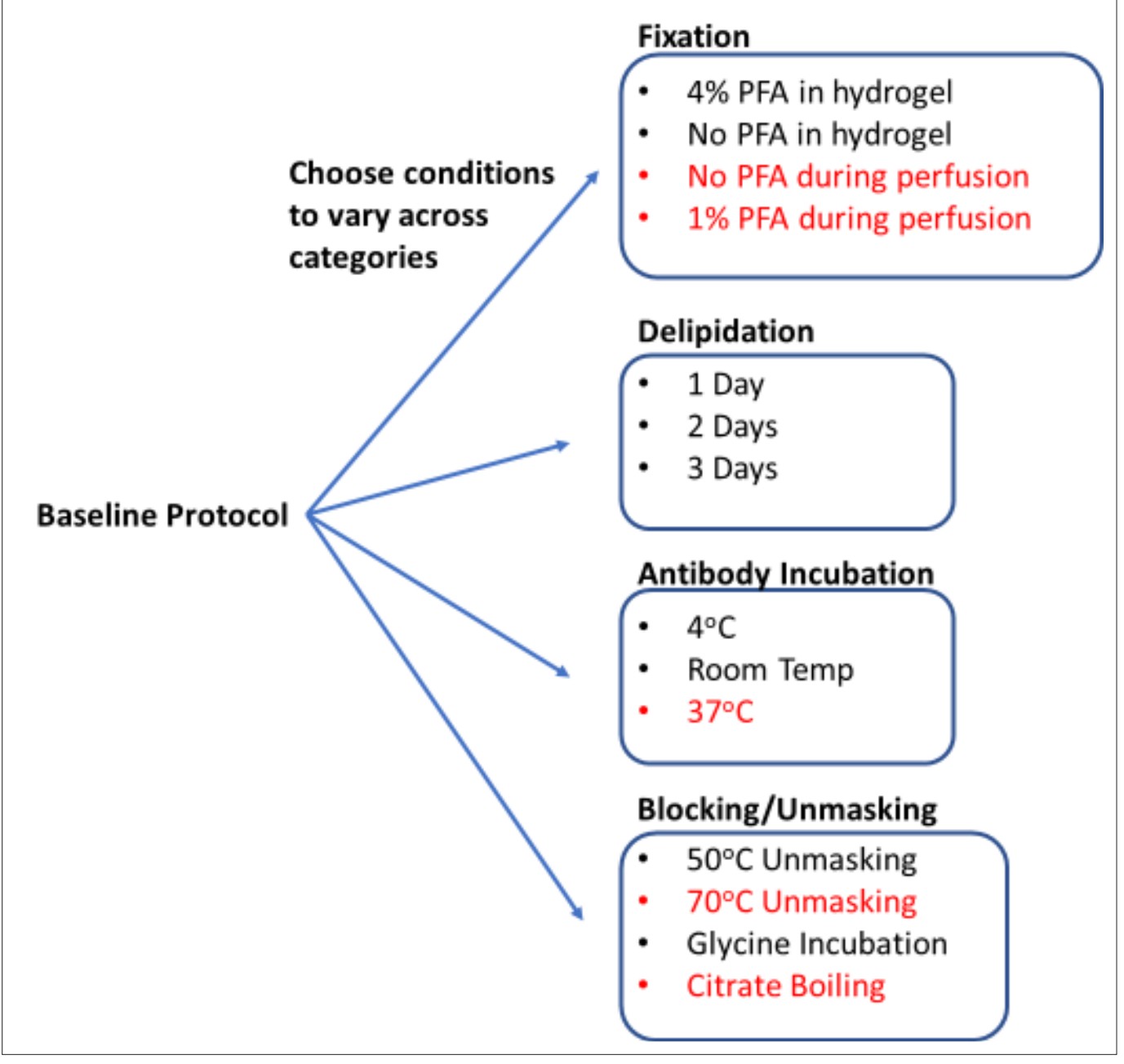

**Figure 1.** Process for generating a library of tissue prepared with different protocols. Start with a baseline protocol (see Materials and methods). Choose conditions to test that can be separated into categories believed to influence staining quality. Prepare tissue by varying only one or two conditions from the baseline at a time. Each protocol should only vary by a single condition from at least one other protocol to allow for a complete understanding of how each change influences staining. So resulting immunohistochemistry (IHC) quality can isolate the effect of each condition change. Stored library of cleared tissues then allows for rapid antibody testing to quickly determine optimal IHC conditions for each antibody as well as protocols that allow for multiplexed IHC. We consistently found conditions in red to have exclusively negative effects on IHC or tissue integrity, so they were excluded from testing with most antibodies.

and it is possible that conditions that reduce the antigenicity may significantly enhance the apparent penetration rate.

This analysis does not require staining of images to be perfect nor precise but requires features that can at least be separated via pixel classification. For images with unintended background staining, Ilastik can be trained to separately classify intended and unintended binding. Note that Imaris is also capable of performing similar pixel classification by integrating Labkit, which we did not compare.

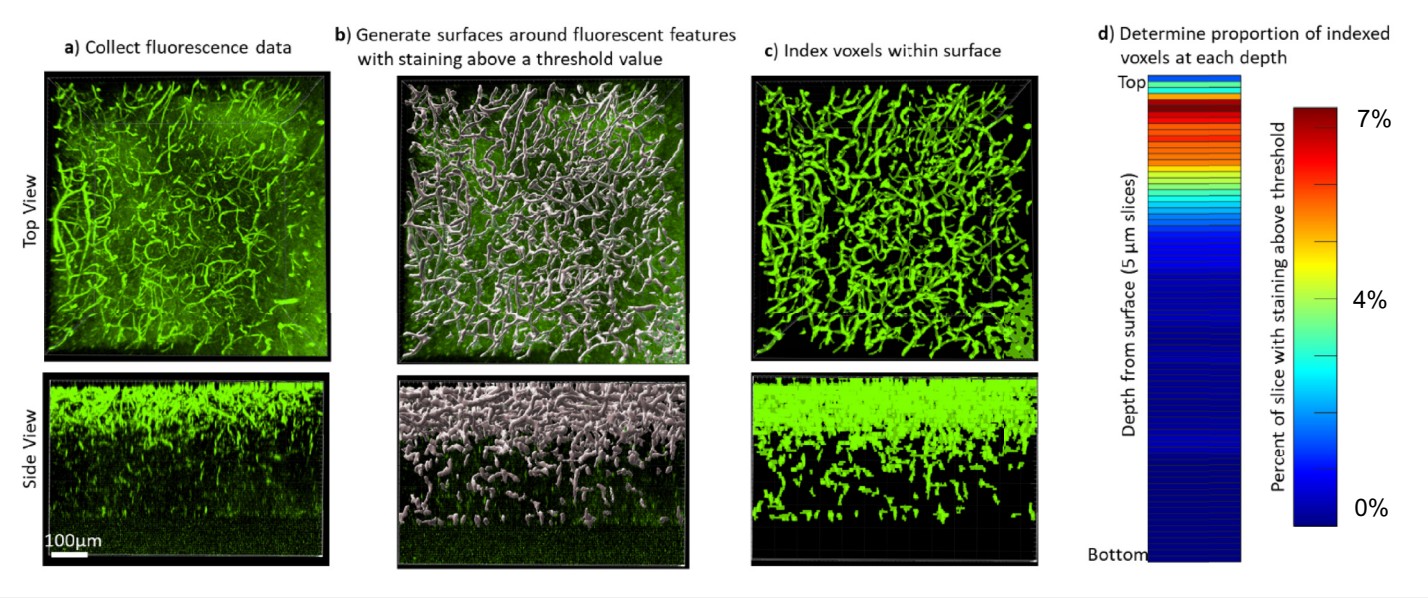

**Figure 2.** Quantifying the staining quality for comparison across protocols and antibodies. Immunohistochemistry (IHC) quality comparisons can be difficult to determine by eye due to differences in fluorescence intensity and background, as well as the many ways that software can modify the visual representation of data, so we created a pipeline to quickly quantify staining quality in a way that allows rapid comparison. (**a**) First, a confocal or lightsheet microscope is used to image a Z-stack throughout the entire thickness of the sample. All imaging parameters are kept identical for a given antibody/fluorophore to allow for comparison between protocols. (**b**) Fluorescence values were normalized across the sample, then a rolling-ball algorithm was used in Imaris software to generate surfaces around fluorescent features with intensity above a threshold value. Alternatively, segmentation masks were generated using Ilastik following their pixel classifier workflow. (**c**) The voxels within the surface were indexed to define the array of voxels with and without staining that could then be analyzed in Matlab. (**d**) The staining quality was quantified and visualized by plotting the ratio of voxels with staining over the total number of voxels in a plane at that depth in the image, the average signal intensity within the segmented region, and the average noise intensity outside of the segmented region (*Figure 3—figure supplements 2 and 3*, *Figure 7—figure supplements 1 and 2*). A higher proportion of indexed voxels, higher signal, lower noise, and longer depth to reach half-max staining intensity all represent better staining quality. This can also be used to provide visual representation of reproducibility (*Figure 3—figure supplements 4–6*). It is also possible to reduce these numbers into a figure of merit that can be quantitatively compared between conditions (*Figure 3c*, *Figure 3—figure supplement 7*).

After acquiring segmentation results, turning it into an ideal figure of merit will vary depending on the antigen of interest and its distribution in tissue. For antigens that have relatively constant distribution over the depth of imaging, the area of each plane within a Z-stack that is segmented could be used as a figure of merit for staining quality across conditions. However, if the antigen distribution does not have a constant distribution, it is better to use a figure of merit that is normalized based on the area of segmentation, such as the average staining intensity at each depth plane.

## Conditions that influence antibody and epitope preservation

Starting from a comprehensive collection of published IHC protocols (*Richardson et al., 2021*; *Kim et al., 2016*; *Scardigli et al., 2021*; *Richardson and Lichtman, 2015*), we selected a subset of commonly varied conditions to create 12 test protocols for proof of principle. Each protocol is nearly identical, only varying by a single condition from at least one other protocol to allow for a complete understanding of how each change influences staining. The baseline protocol involved minimal processing and gentle conditions. Specifically, the tissue was fixed in 4% paraformaldehyde (PFA) for 2 days post-perfusion, embedded in hydrogel made of 4% acrylamide, delipidated in sodium dodecyl sulfate (SDS) for 1 day at 37°C, rinsed in phosphate-buffered saline (PBS) with 0.2% Triton X-100 (PBST) at 4°C, and incubated with antibodies at 4°C for 18 hr. We then created a library of tissue that was processed using small changes to this baseline protocol, so we could compare how each of these changes influenced IHC quality. This library included changes in the composition of the hydrogel (with and without 4% PFA), changes in delipidation time (1, 2, 3 days), additional steps before antibody incubation (such as unmasking with 50°C or incubation with glycine), and differential changes in antibody incubation temperature (4°C or room temperature).

In addition to this library, we also examined the effects of boiling samples in citrate buffer before or after delipidation, heating to temperatures above 70°C before immunolabeling, and removal or decrease of PFA concentration in the transcardial perfusion or post-perfusion fixing step (*Figure 1*). However, these conditions were universally observed to negatively impact IHC or tissue integrity, so they were excluded from our final library.

We chose two antibodies commonly used in neuroscience studies of cleared volumes (Glut1 and NeuN) and performed IHC on the same tissue sections from a mouse cortex that were processed according to the 12 protocols (*Figure 1*, *Figure 3*). Each antibody was directly conjugated to a fluorophore: Glut1 to Alexa Fluor 488 and NeuN to Alexa Fluor 568. Increasing delipidation time to 2 days had a positive influence on NeuN staining, and no influence on Glut1 staining, but both showed worse IHC when increasing the delipidation time further. Incubation of tissue at 50°C for 1 hr prior to antibody incubation improved staining for Glut1 and NeuN. Glycine treatment had no significant effect on NeuN staining, but inhibited Glut1 staining. Incubating antibodies at room temperature improved NeuN staining and on Glut1 staining. Intriguingly, staining with DAPI appears insensitive to the different protocols (*Figure 4*). Inclusion of PFA in the hydrogel resulted in decreased IHC quality for both Glut1 and NeuN, however this significantly improved IHC of dopamine receptor 2 (*Figure 4—figure supplement 1*). Therefore, we ascertained the optimal protocol for simultaneously staining Glut1, NeuN, and DAPI was to delipidate for 2 days, place the tissue at 50°C for 1 hr before incubating with antibodies, and incubating the antibodies at room temperature. Importantly, these same conditions would result in no staining of dopamine receptor 2 (*Figure 4—figure supplement 1*). Instead, an IHC protocol is required to use PFA in the hydrogel step to show adequate staining of dopamine receptor 2, which can still allow for simultaneous staining of NeuN, Glut1, and DAPI if the tissue is only delipidated for 1 day and incubated with antibodies at 4°C. Together, these results show that different conditions can have opposite effects on the quality of staining and depth of antibody penetration and reinforce the importance of optimizing sample preparation conditions for each antibody.

## Comparison of IHC conditions for different antibodies directed against a single protein

A common strategy following unsuccessful IHC in tissue is to test antibodies from different suppliers or antibodies raised against different epitopes in the desired target. To better understand whether optimal IHC protocols may be shared for antibodies against the same target, we compared the staining of a monoclonal GFAP directly conjugated to a fluorophore and a polyclonal GFAP that was detected with a fluorophore conjugated to a secondary antibody. To enable a direct comparison, both sets of antibodies were used to label the same tissue sections. Immunolabeling GFAP with different antibodies shows that the trends in staining quality are similar for both antibodies across all conditions (*Figure 5*). This suggests that conditions that enhance GFAP stability and prevent denaturation of its epitopes have a positive effect on all GFAP antibodies provided the antibodies are of high quality, have high specificity and limited off-target binding. This also suggests that IHC of GFAP with these antibodies is limited more by denaturation of the epitope than the antibody itself. We found that both GFAP antibodies successfully labeled their target under similar conditions (*Figure 5*), and both showed similar conditions where no labeling occurred (*Figure 5—figure supplement 1*). Some differences in labeling quality are apparent between antibodies, however it is not clear whether the differences in labeling quality are due to the conjugation of the fluorophore or other differences between the monoclonal and polyclonal antibodies. As this approach also reduces the length of the staining protocol, it is recommended to use primary antibodies directly conjugated to fluorophores whenever possible.

Multiple antibodies that recognize tau protein were also compared including AT8 anti-tau, HT7 anti-tau, and DAKO total tau antibodies. Tau is a microtubule associated protein that is primarily expressed by neurons and is the main component of neurofibrillary tangles in Alzheimer's disease (AD). We chose three commonly used antibodies against tau and tested them first in tissue from a transgenic mouse model, hTau (*Andorfer et al., 2003*), that expresses normal human tau isoforms.

There was significantly more variability between tau antibodies than seen with GFAP antibodies, primarily due to the inability for DAKO total tau antibody to effectively penetrate beyond the tissue's surface (*Figure 6*, *Figure 6—figure supplement 1*). However, as the DAKO total tau antibody was

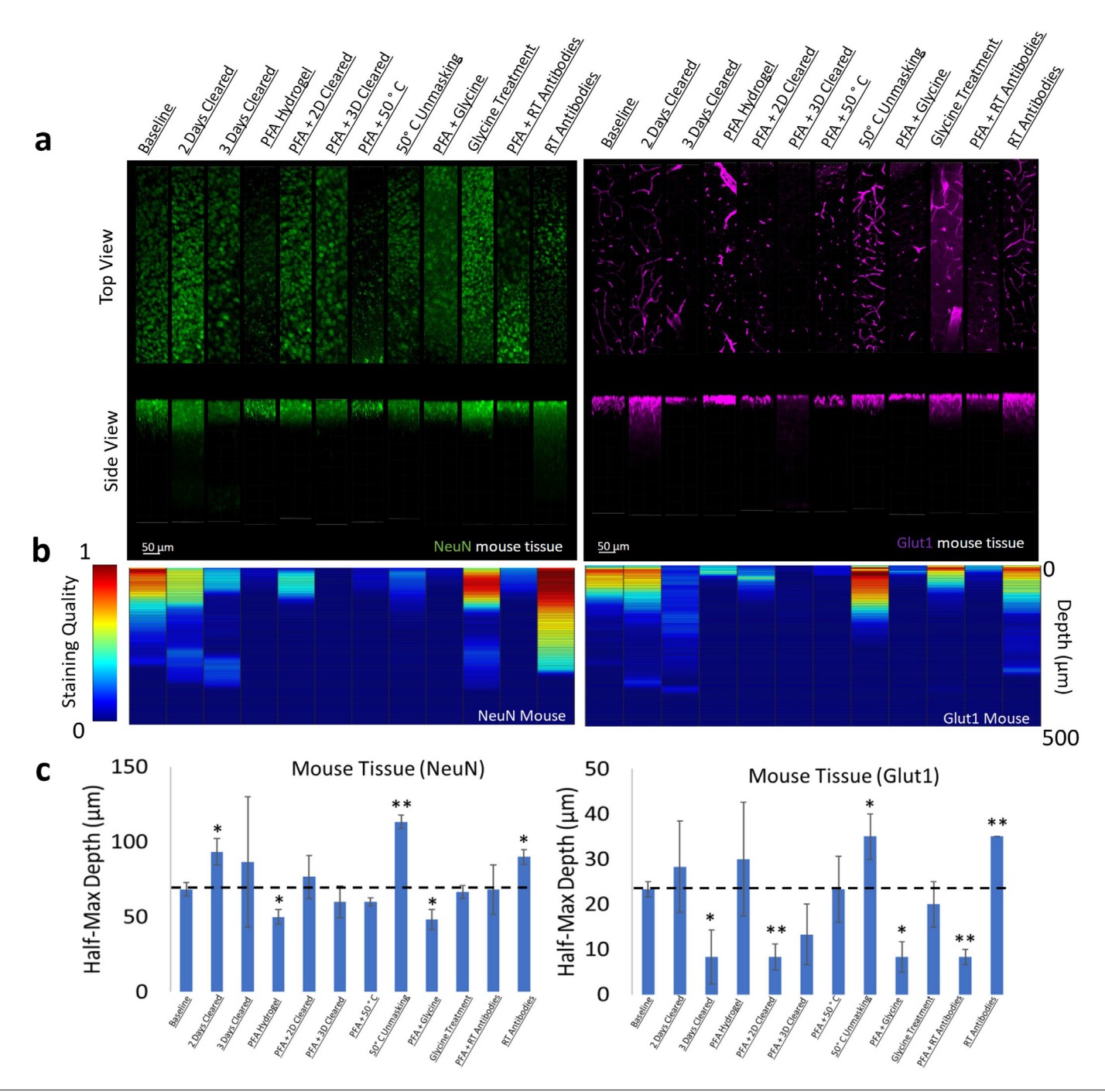

**Figure 3.** Influence of common conditions on immunohistochemistry in mouse brain tissue. (**a**) Representative maximum intensity projection views from the top and side of 500 µm thick, 5 mm wide tissue sections. C57Bl/6J mouse isocortex was processed, stained, and imaged simultaneously with NeuN (green, left panels) and Glut1 (purple, right panels). (**b**) Quantitative comparison of immunohistochemistry labeling across conditions using Imaris. A rolling-ball subtraction was used to determine the area with signal-to-noise above cut-off for every image plane in the Z-stack. The proportion of indexed voxels in each plane was then divided by the maximum value across all conditions to generate a normalized staining quality score, which allows for quick visual assessment of staining quality differences with depth and between conditions. (**c**) Quantitative comparison of the depth at which the immunohistochemistry (IHC) staining intensity becomes half of the maximum, using segmentation results from Ilastik. Error bars represent standard error. Two-tailed t-test comparing results to baseline were represented with $p<0.05$ = *, $p<0.01$ = **, and no significance = unlabeled. N=3 for all samples.

The online version of this article includes the following figure supplement(s) for figure 3:

*Figure 3 continued on next page*

*Figure 3 continued*

**Figure supplement 4.** Reproducibility of NeuN immunohistochemistry and analysis in mouse brain tissue.

**Figure supplement 5.** Reproducibility of Glut1 immunohistochemistry and analysis in mouse brain tissue.

**Figure supplement 6.** Reproducibility of NeuN and Glut1 immunohistochemistry corrected signal intensity in mouse brain tissue.

**Figure supplement 7.** Reducing comparison of total stained volume to one dimension.

**Figure supplement 8.** Immhunohistochemistry staining consistency across brain regions.

not directly conjugated to a fluorophore, we cannot exclude the possibility that the inability to image DAKO-stained tau beyond the surface may be an effect of poor penetration and diffusion of the secondary antibody. Importantly, when used in traditional IHC in microtome slices of human tissue, all three of these antibodies labeled similar structures (*Figure 6—figure supplement 2*). As expected, similar trends in optimal IHC protocols were observed for AT8 and HT7 anti-tau antibodies in mouse

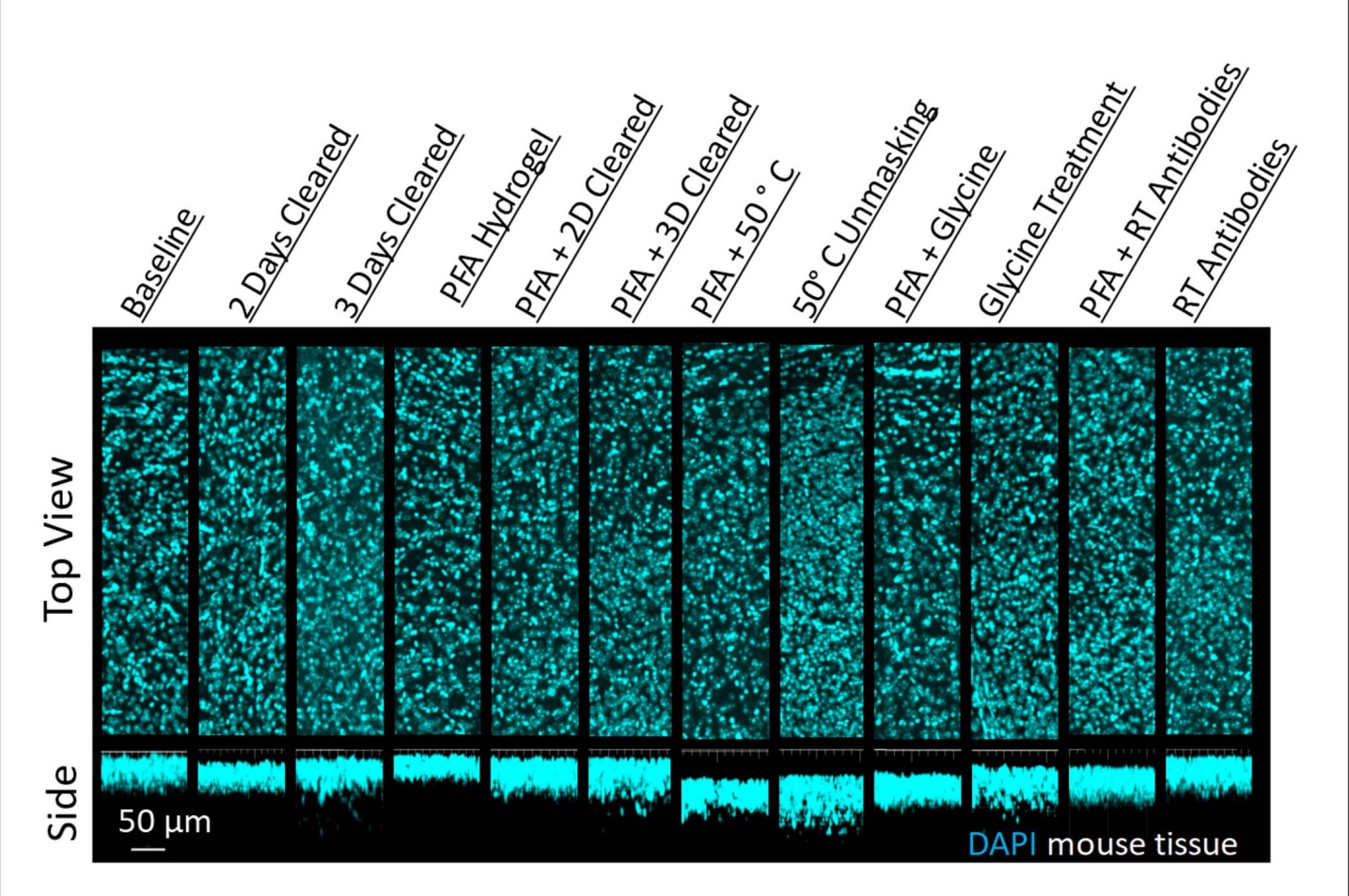

**Figure 4.** Influence of common conditions on 4',6-diamidino-2-phenylindole (DAPI) staining. Representative maximum intensity projection views from the top and side of 500 µm thick, 5 mm wide tissue sections of C57Bl/6J mouse isocortex stained with DAPI.

The online version of this article includes the following figure supplement(s) for figure 4:

**Figure supplement 1.** Immunohistochemistry of dopamine receptor 2 in mouse brain tissue.

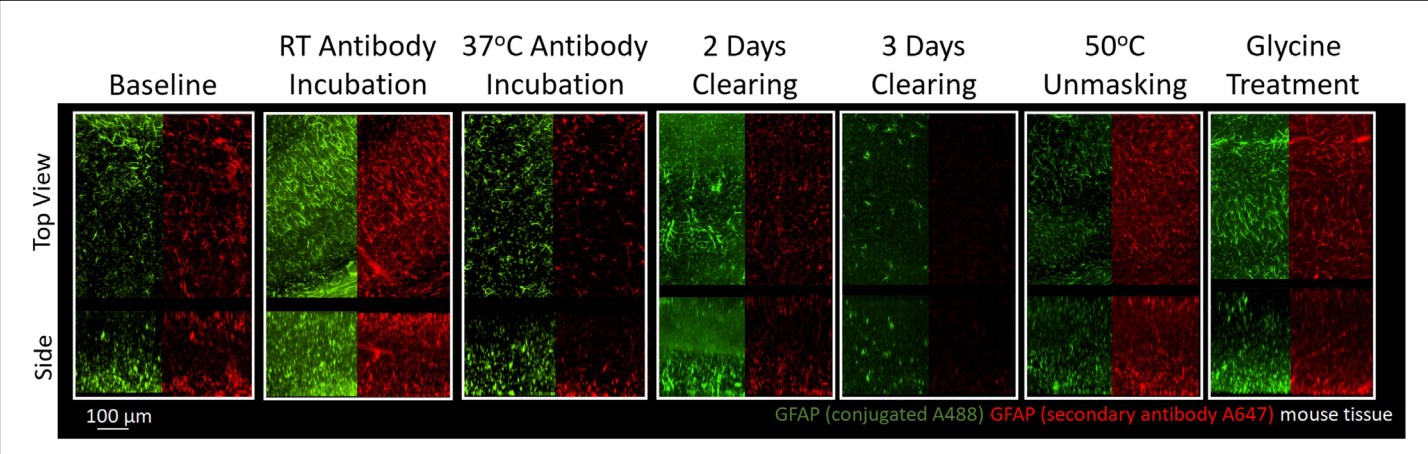

**Figure 5.** Immunohistochemistry of two different antibodies for GFAP. Representative maximum intensity projection views from the top and side of 500 μm thick, 5 mm wide tissue sections of C57Bl/6J mouse isocortex stained with two different antibodies for GFAP. Monoclonal GA5 conjugated to Alexa Fluor 488 (green) was stained and imaged in the same tissue as a polyclonal antibody using a goat anti-rabbit secondary conjugated to Alexa Fluor 647 (red). Other protocols show no significant labeling (*Figure 5—figure supplement 1*).

The online version of this article includes the following figure supplement(s) for figure 5:

**Figure supplement 1.** Immunohistochemistry with GFAP antibodies in with paraformaldehyde (PFA)-hydrogelled mouse brain tissue.

tissue expressing human tau (*Figure 6—figure supplement 3*) and showed no labeling in a negative control; mice without tau expression (*Figure 6—figure supplement 4*).

## Mouse brain tissue as a model system for optimizing IHC in cleared human tissue

Optimizing IHC in human tissue is especially difficult due to the limited amount of tissue available for experimentation. Therefore, we performed IHC optimization in both mouse and human brain tissue for individual antibodies and analyzed their similarities. Surprisingly, our testing of IHC protocols for NeuN and Glut1 in human tissue showed similar changes in staining quality and depth when compared to mouse tissue (*Figure 7*, *Figure 7—figure supplements 1 and 2*). Both human and mouse tissue showed decreases in staining quality with increasing clearing time and fixation with PFA. Room temperature antibody incubation as well as epitope unmasking at 50°C for 1 hr prior to antibody incubation improved Glut1 and NeuN staining in both mouse and human tissue. It is also worth noting that this NeuN antibody appears to accumulate off-target in the human tissue, which is not observed in the mouse tissue.

The optimal protocol that allows NeuN, Glut1, and AT8-tau staining was then determined by finding the conditions that allow for the simultaneous IHC of all three antibodies in a single tissue section, and then finding the combination of conditions among those that maximizing the staining quality. It is worth noting that this was not necessarily the best protocol for each individual antibody. It is more important to have adequate staining of each epitope than to have fantastic staining of one at the cost of another. As described above, we determined the optimal protocol for multiplexed labeling by combining the protocol changes that quantitatively improve the staining quality, defined by half-max depth, for each antibody above the baseline protocol and do not significantly degrade performance of other antibodies. For NeuN, Glut1, and AT8 antibodies this protocol involves clearing for 2 days, using 50°C to unmask epitopes, and incubation of antibodies at room temperature. The optimal protocol was then validated in 1 mm thick pieces of human brain tissue, approximately 100-fold thicker than typical paraffin embedded sections. Antibodies for tau in mice expressing human tau show similar variability across conditions for the labeling of labeling tau in human tissue (*Figure 6*). We concurrently incubated the human tissue with three antibodies conjugated to three different fluorophores (NeuN-A568, Glut1-A488, and AT8-A647). These antibodies were used to stain tissue that was first embedded in a 4% acrylamide hydrogel, cleared for 2 days and thoroughly washed. The antibodies were incubated with the tissue for 14 days to ensure complete staining through the whole sample in both humans (*Figure 8*) and mice (*Figure 8—figure supplement 1*). It should be noted

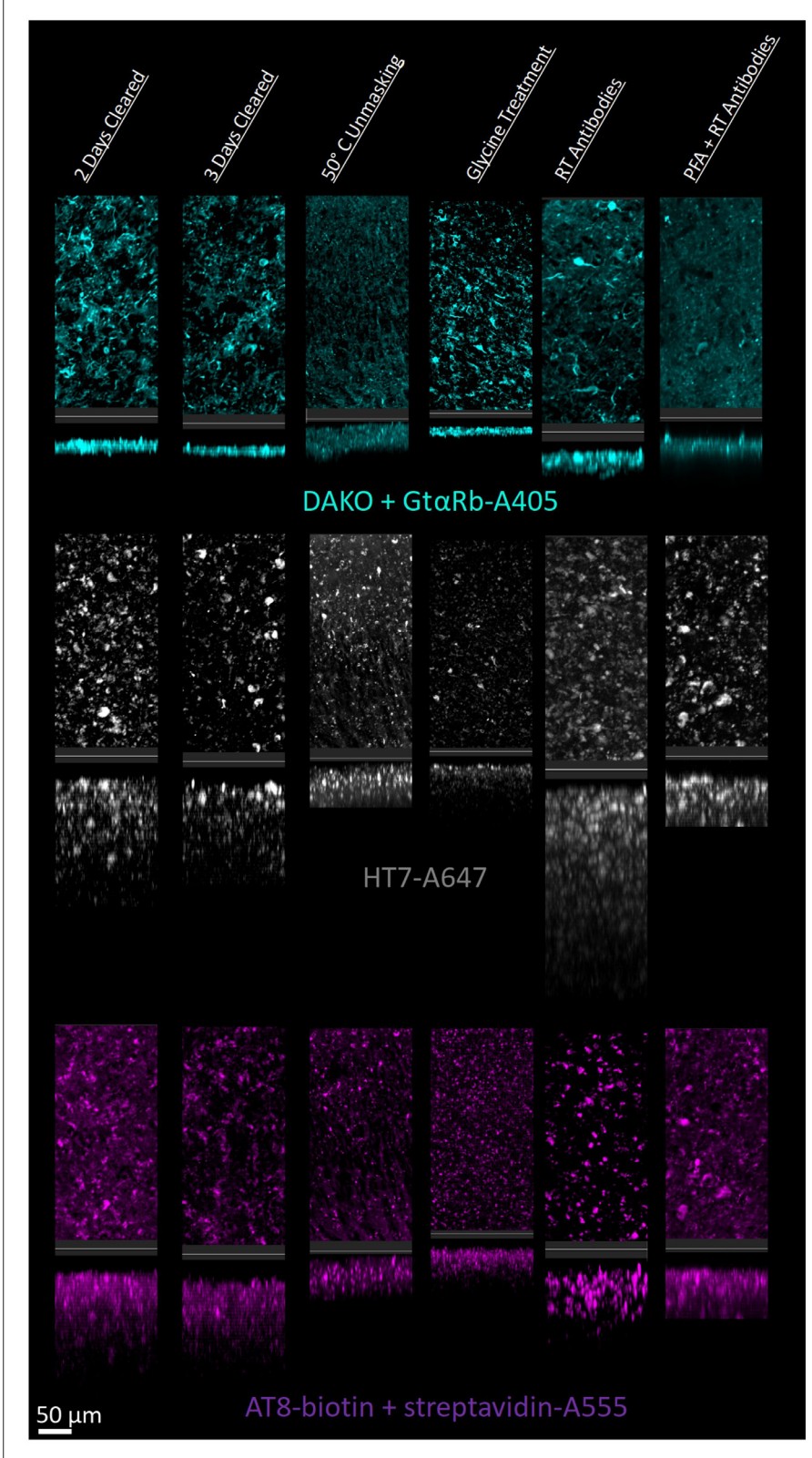

**Figure 6.** Tau immunohistochemistry in cleared human tissue. Representative maximum intensity projection views from the top and side of 500 µm thick, 5 mm wide tissue sections. Other protocols showed worse staining quality (*Figure 6—figure supplement 2*).

*Figure 6 continued on next page*

*Figure 6 continued*

The online version of this article includes the following figure supplement(s) for figure 6:

**Figure supplement 1.** Additional Dako, AT8, and HT7 tau immunohistochemistry conditions in human brain tissue.

**Figure supplement 2.** Representative microtome section images taken from the human tissue prior to clearing.

**Figure supplement 3.** AT8 and HT7 immunohistochemistry in mouse tissue expressing human tau (Htau).

**Figure supplement 4.** AT8 and HT7 immunohistochemistry in tau knockout mice (Htau KO).

that we found 3 days of antibody incubation sufficient for consistent and thorough staining across brain regions in 500 µm mouse tissue (*Figure 3—figure supplements 8 and 9*), and there were no notable downsides to leaving the antibody incubating longer. By carefully following the optimal IHC protocol determined in this work, we were able to achieve complete penetration and simultaneous labeling of the 1 mm thick human tissue with all three antibodies. Both the Glut1-labeled vasculature and AT8-labeled tau are extremely well defined and could be easily quantitated. The NeuN reflects the same quality that it had during the troubleshooting but does not appear to be an ideal antibody for assessment of human neurons under these conditions.

## Discussion

Although it is well known that alterations to IHC protocols can have detrimental effects to the staining results, many researchers accept the dogma that a universal IHC protocol exists that is robust enough to support labeling with a wide range of antibodies. When a subset of antibodies fail in a multi-target IHC protocol, often the antibodies themselves are blamed. Here, we have found that small changes to an IHC protocol can have profound effects on labeling efficiency. Rather than the antibody, our data suggests that poor staining is most often due to denaturation or blockage of the epitope. These procedural differences do not influence every antibody-antigen pair in the same way, so the optimal protocols for IHC should be determined for each individual target. Using combinations of antibodies in the same tissue may require using a protocol with conditions that are acceptable for all antibodies, rather than conditions that are optimal for each individual antibody. Unfortunately, as we found with the dopamine receptor 2 (*Figure 4—figure supplement 1*) and GFAP (*Figure 5*, *Figure 5—figure supplement 1*) IHC, a protocol may not exist to allow for both targets to be effectively stained simultaneously. However, the methods described here allow a researcher to rapidly determine (1) an optimal protocol for single antibody staining, (2) an acceptable protocol that allows simultaneous multi-antibody staining, or (3) that cyclical single-antibody staining and stripping will be required to match conditions necessary for each antibody.

Importantly, temperature, fixation conditions, and the other conditions investigated here have greater influence, the larger the tissue volume becomes. For many of the tested protocols, the difference in staining is not apparent at the surface which suggests that the use of tissue sections that are too thin does not allow for appropriate optimization. Many of the protocols we tested for NeuN or Glut1 produce some degree of labeling at the surface of the sample that would have appeared no different between samples thinner than 50 µm, despite obvious differences in staining quality at depth.

Most surprising to us was that mouse tissue is a good model system for determining large-volume IHC protocols for human samples. However, these results show that minor procedural changes can disrupt the quality of staining, so it is important that tissue preparations are identical.

IHC protocols in large-volume tissue that requires clearing can take weeks or months to complete depending on the initial state and size of the tissue (*Susaki et al., 2020*; *Ueda et al., 2020a*). Previous studies have suggested that human tissue requires processing for longer than comparable-sized rodent tissue. Our results suggest that this is not due to intrinsic differences between human and mouse brain tissue, but rather a result of differences in the state of the tissue when clearing experiments are started. Human tissue is often stored in fixative for extended periods of time and, as our data shows, even a small increase in PFA exposure can have significant influence on IHC quality and depth. To our surprise, tissue that was delipidated for 3 days showed less rapid IHC penetration than tissue that was cleared for 2 days in all cases except for DAPI labeling. This suggests that samples can

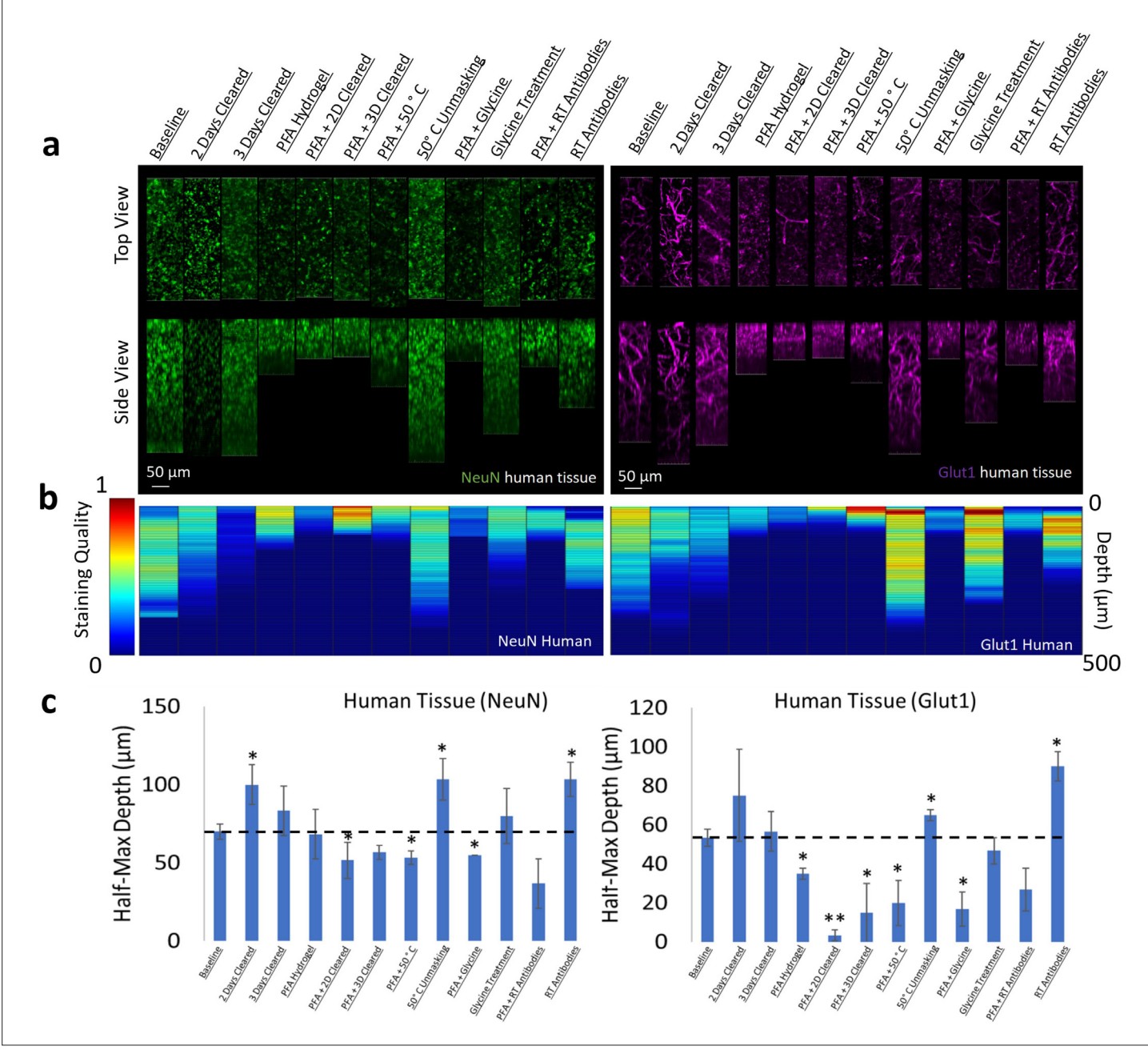

**Figure 7.** Influence of common conditions on immunohistochemistry in human brain tissue. (**a**) Representative maximum intensity projection views from the top and side of 500 µm thick, 5 mm wide tissue sections. Tissue was processed, stained, and imaged simultaneously with NeuN (green, left panels) and Glut1 (purple, right panels). (**b**) Quantitative comparison of immunohistochemistry labeling across conditions using Imaris. A rolling-ball subtraction was used to determine the area with signal-to-noise above cut-off for every image plane in the Z-stack. The area in each plane was then divided by the maximum value across all conditions to provide a staining quality score, which allows for quick visual assessment of staining quality differences with depth and between conditions. (**c**) Quantitative comparison of the depth at which the immunohistochemistry (IHC) staining intensity becomes half of the maximum, using segmentation results from Ilastik. Error bars represent standard error. Two-tailed t-test comparing results to baseline were represented with $p < 0.05$ = *, $p < 0.01$ = **, and no significance = unlabeled. N=3 for all samples.

The online version of this article includes the following figure supplement(s) for figure 7:

**Figure supplement 1.** Immunohistochemistry performance of NeuN in human tissue.

**Figure supplement 2.** Immunohistochemistry performance of Glut1 in human tissue.

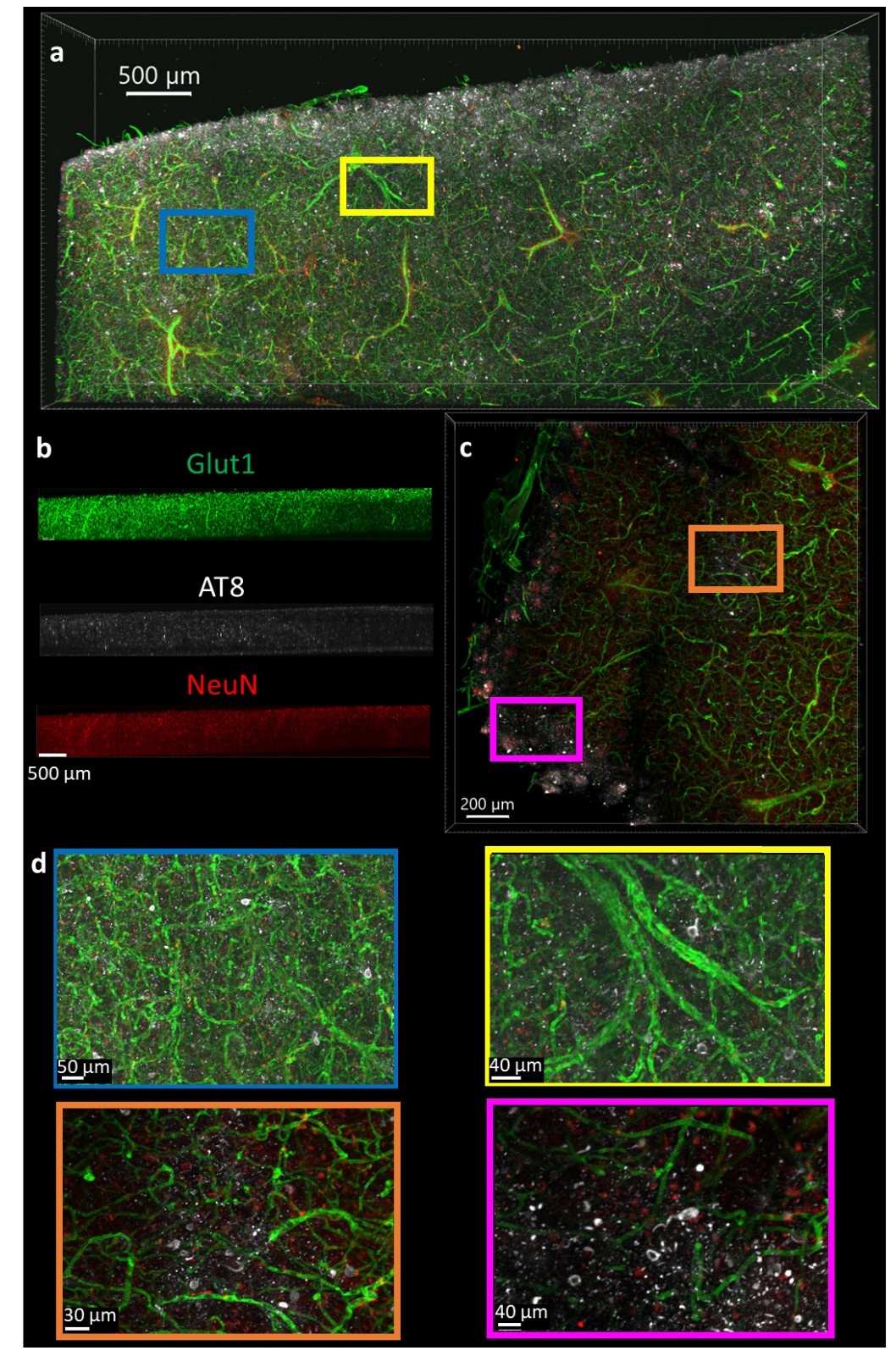

**Figure 8.** Immunohistochemistry in 1 mm thick human tissue. The tissue was crosslinked without paraformaldehyde (PFA), cleared for 2 days, rinsed, then the antibodies NeuN-A568 (red), Glut1-A488 (green), and AT8-A647 (white) were incubated with the tissue for 14 days. (**a**) Top-down view of fluorescence from 1 mm section human brain. (**b**) Side view of each antibody fluorescence through entire section depth. (**c**) A 200 µm virtual section

*Figure 8 continued on next page*

*Figure 8 continued*

of human tissue, from a 1 mm thick sample. (**d**) Zoomed-in regions from the full images show that AT8-labeled tau tangles are present at multiple locations throughout the human brain sections.

The online version of this article includes the following figure supplement(s) for figure 8:

**Figure supplement 1.** Immunohistochemistry in 1 mm mouse tissue.

quickly become over-cleared and lose antigenicity which may extend the amount of time previously reported as necessary for IHC. It follows that future experiments should explore labeling tissue with antibodies with even shorter or milder fixation conditions and, where possible, carried out prior to any tissue clearing procedures.

Our methodology enables rapid optimization of IHC protocols, including protocols that require tissue clearing and provides a simple path to imaging tissue blocks that are 100 times thicker than conventional tissue slices. First, a library of cleared tissue samples that were each processed using different conditions is generated that can be stored for future experiments. Then, optimal IHC conditions are determined for new antibodies within a few days (approximately 24 hr for staining, 24 hr for rinsing, 12–24 hr for refractive index matching/clearing, 3 hr to image the samples, and a similar amount of time to quantify and compare the staining quality). The variability in staining quality across protocols suggests that researchers should use this process to test how changes may affect antibodies or epitopes that they plan to study before committing many weeks or months to a procedure in larger volumes. Additionally, these results speak to the necessity for scientists to publish both conditions that improve and degrade specific antibody performance. Understanding which procedures and antibodies are appropriate for a given sample will be essential for tissue clearing techniques to gain widespread use in human tissue. We are optimistic that our optimization process will allow researchers use these techniques to successfully label cleared human tissue, which will improve the adoption of this technique and the ability to unlock new insight into human pathology and neuroanatomy.

# Materials and methods

## Human tissue samples

Human tissue was provided by the Massachusetts Alzheimer's Disease Research Center (ADRC) with approval from the Mass General Brigham IRB (1999P009556). Three human participants with AD were selected from the Massachusetts Alzheimer's Disease Research Center. Sex, age at death, Braak staging, which hemisphere, and comorbidities are listed in *Supplementary file 1*. Autopsy tissue from human brains were collected at Massachusetts General Hospital, with informed consent of patients or their relatives and approval of local institutional review boards. Human tissue was received unfrozen immediately after autopsy and fixed (sample 2620) or frozen immediately after autopsy then thawed prior to fixation (samples 1581 and 1762). Tissue was placed in 4% PFA for 48 hr at 4°C, transferred to PBS for 24 hr at 4°C before slicing. This tissue was treated identically to the mouse tissue described below in the tissue slicing procedure and onward.

## Mice strains

Male and female C57Bl/6J mice (Jackson Laboratory 000664) were used for most tissue experiments. For tissue to test IHC of tau, 10- to 12-month-old male hTau mice bred from a local colony (B6.Cg-Mapt$^{tm1(EGFP)Klt}$ Tg(MAPT)8cPdav/J, Jackson Laboratory 005491) that express only human tau isoforms were used. hTau knockout mice bred from a local colony (Mapt$^{tm1(EGFP)Klt}$/J, Jackson Laboratory 004779) were used as controls for tau IHC experiments. Mouse experiments were performed with approval of local institutional review boards (IACUC protocol # 2019N000026).

## Perfusion

Mice were transcardially perfused with ice-cold 40 ml PBS, followed by 40 ml 4% PFA (Electron Microscopy Sciences). Brains were removed from the cranium and placed in 4% PFA for 48 hr at 4°C. Brains were then transferred to PBS for 24 hr at 4°C before slicing.

## Tissue slicing

Brains were embedded in 4% agarose hydrogel. To do this, extracted brains were placed individually in a small Petri dish and let dry slightly. The agarose solution was dissolved by boiling in a microwave, then left on the counter to cool before pouring. Agarose was poured to completely cover the top of the brain, which allows it to provide a solid support to prevent deformation during cutting. A block of agarose containing the brain was placed in a vibratome (VT1000S vibrating blade microtome, Leica, Wetzlar, Germany) then cut into 500 or 1000 µm coronal or sagittal slices. Each slice was then removed from the agarose through gentle manipulation with blunt forceps or paintbrushes before crosslinking.

## Tissue hydrogel crosslinking

Each slice was incubated with a hydrogel crosslinking solution for 1 day at 4°C to let the molecules diffuse through the tissue. Solutions should be kept cold before and after the addition of VA-044 to prevent premature initiation of polymerization. Different crosslinking solutions were used to determine the effect of including PFA on IHC. *Crosslinking solution 1*: PBS with 4% (wt/vol) acrylamide (Sigma-Aldrich, St Louis, MO) and 0.25% (wt/vol) VA-044 thermal polymerization initiator (Thermo Fisher Scientific, Pittsburgh, PA). *Crosslinking solution 2*: PBS with 4% (wt/vol) acrylamide, 0.25% (wt/vol) VA-044 thermal polymerization initiator, and 4% PFA. After incubation with the crosslinking solution for 1 day, the tissue was left in solution and placed in vacuum at 37°C for 3 hr. After polymerization with X-CLARITY polymerization system (Logos Biosystems, South Korea), the slices were rinsed with 50 ml PBS three times over 3 hr. Each slice was then cut into small pieces with width and length at least twice the thickness. This was done to increase the number of conditions that could be tested with small amounts of tissue.

## Delipidation

After hydrogel crosslinking, the tissue was placed in a delipidation solution (SDS 200 mM, sodium sulfite 20 mM, sodium borate 20 mM, sodium hydroxide to pH 8.5–9, filtered if necessary) at 37°C for different amounts of time (1, 2, or 3 days), to determine the effect of delipidation in IHC. After delipidation, the brain slices were rinsed with 50 ml PBST (PBS with 0.2% Triton X-100, Thermo Fisher Scientific) five times over 1 day.

## Pre-IHC tissue processing

Different conditions were tested at this step to determine their influence on IHC. Each tissue section was tested using only one of the following conditions. *Baseline:* Tissue was left at 4°C in PBST. *50°C Condition:* Tissue was heated to 50°C for 1 hr in PBST. *Glycine:* Tissue was incubated with 0.1 M glycine at 4°C for 12–18 hr. *Glycerol:* Tissue was placed in a low osmolarity solution of 80% glycerol/20% water for 12–18 hr. *Blocking:* Tissue was placed in 5% BSA in PBS for 12–18 hr.

## Storage of tissue library

The tissue library was then used immediately or stored at 4°C in PBS with 0.02% (wt/vol) sodium azide. Mouse and human tissue was processed following all of the steps above and then used up to 3 months later. Additional mouse and human tissue was stored following delipidation, then successfully used up to 6 months later by resuming from the pre-IHC tissue processing state. IHC conditions for a given antibody were always compared using batches of tissue that were stored for the same amount of time.

## Immunohistochemistry

- The following antibodies were purchased then stored according to the vendors specifications: NeuN antibody conjugated to Alexa Fluor 568 (Abcam, Ab207282).
- GFAP antibody unconjugated (Abcam, ab16997) and conjugated to Alexa Fluor 488 (Thermo Fisher, 53-9892-82).
- Glut1 antibody conjugated to Alexa Fluor 488 (EMD Millipore, 07-1401-AF488).
- Phospho-tau antibody AT8 conjugated to biotin (Thermo Fisher, MN1020B).
- Tau monoclonal antibody HT7 conjugated to Alexa Fluor 647 (Thermo Fisher, 51-5916-42).
- DAKO total tau antibody unconjugated (Agilent, A0024).
- Dopamine receptor 2 antibody (Santa Cruz Biotechnology, SC5303)

- Secondary antibody Goat Anti-Rabbit IgG H&L conjugated to Alexa Fluor 405 (Abcam, ab175652) and Alexa Fluor 647 (ab150079).
- Streptavidin conjugated to Alexa Fluor 568 (Thermo Fisher, S11226).

Each brain slice was placed in a 2 ml Eppendorf tube (or similar vessel) that could hold the slice so its large, flat sides could be exposed to solution. PBST was added to just cover the top of the samples (~300 µl). One to 5 µl of stock antibody solution was added, depending on the concentration of the stock. The final concentration of each antibody in solution was 5 µg/ml. Fluorophores or host species were chosen so multiple different antibodies could be tested at the same time in a single slice of tissue. The tissue was then incubated with the antibodies at 4°C or room temperature for 18 hr. The tissue was then rinsed in PBST 3×50 ml overnight (18–24 hr) at room temperature. If using unconjugated primary antibodies, the secondary antibodies were then added to give a final concentration of 5 µg/ml and incubated for 48 hr at 4°C. The tissue was then rinsed in PBST 3×50 ml overnight at room temperature.

## IHC of human microtome sections

Prior to preparation of cleared tissue volumes, fresh/frozen human samples were tested for the presence of AT8, HT7, and DAKO. Microtome sections were fixed in 4% PFA for 15 min, rinsed in Tris-buffered saline (TBS) three times, then blocked in TBS with 5% bovine serum albumin. Primary antibodies were incubated at a concentration of 1:500 overnight. Tissue was rinsed in TBS 3×, then incubated in secondaries at room temperature for 60 min, and rinsed in TBS 3×. Immumount was then used to seal the tissue prior to imaging.

## Refractive index matching

The samples were incubated with 80% glycerol, 15% DI water, 5% PBS (RI-matching solution) for at least 3 hr at room temperature with gentle shaking. The RI-matching solution was then replaced with fresh solution and left to incubate for at least 3 more hr before imaging. Samples were stable stored in this solution at room temperature and could be re-imaged for at least 3 months.

## Imaging

Samples were imaged using Olympus Inverted Confocal FV3000 with a 10× objective except for samples using dopamine receptor antibodies which were imaged using the Zeiss Lightsheet Z.1. Image Z-stacks were then reconstructed and visualized using Imaris microscopy image analysis software.

## Post-imaging analysis

Prior to analysis, images were assigned a random number so the analyzing scientist was blind to tissue clearing conditions but still aware of the antibodies used in the tissue, which was necessary to identify staining.

## Analysis with Imaris

Imaris software was used to assess the staining quality score. First, the distribution of fluorescence values across all slices in the Z-stack were normalized. Then, a rolling-ball algorithm was applied to define the edges of fluorescent shapes with signal intensity greater than background by a threshold value. The area within each slice of a Z-stack that contained staining above this threshold was then divided by the maximum across all tissue stained with that fluorophore to determine the staining quality throughout the tissue. This metric was only applied to stains like NeuN or Glut1 in the cortex where we can make the assumption that there is a uniform and consistent distribution of these cells across sections. All samples that were imaged as part of the library described in *Figure 1* were included in this study, with at least three independent samples per condition to determine reproducibility.

## Analysis with Ilastik

Raw data was converted to HDF5 format using ilastik's ImageJ plugin plus custom macro to batch convert. The staining was then segmented using their pixel classifier workflow (*Berg et al., 2019*). In short, a paintbrush was used to draw over the signal and background to help train the classifier on how to segment each image. Real-time segmentation results were then improved using a minimum of 600 image planes from six different staining conditions that reflect the distribution of staining

quality in the entire set. All images were then batch processed through the trained pixel classifier and exported as multipage tiff.

## Post-image segmentation analysis in Matlab

The image segmentation output from either Imaris or Matlab was saved as a single-channel multipage Tiff and placed into a folder, and all the raw images were saved as a single-channel multipage Tiff and placed into a different folder. Image conversion was done using ImageJ macros. Quantification and generation of data for plotting can then be done by specifying the data path, ensuring the Matlab code is on the path, then running the code. The main code 'QuantifyIHC.mat' should be run first, which goes through the files in each folder in order and determines the segmentation area, staining within the segmentation, and noise outside of the segmentation. Next, running the 'FindHalfHax.mat' code will find the starting plane for plotting, the half-max depth, place all data for plotting into arrays, allow for the re-ordering of blinded data, and plotting of the heatmaps.

## Code availability

All Matlab code and ImageJ macros used to quantify data can be accessed at https://github.com/tjzwang/IHC, (copy archived at swh:1:rev:64acafda08ebdda1021edfe47e3d328b21218aa6; *Zwang, 2022*).

## Acknowledgements

This work was supported by the NIH/NIA-K99AG068602 (TJZ) and the Harrison Gardner, Jr Award (TJZ).

We would like to thank Massachusetts General Hospital Center for comparative Medicine and Analiese Fernandes for help with animal husbandry and care. We would also like to thank the Harvard Center for Biological Imaging including Christian Hellriegal for the use of their imaging facility and helpful conversations. Autopsy tissue from human brains were collected at Massachusetts General hospital, with informed consent of patients or their relatives and approval of local institutional review boards.

## Additional information

### Competing interests

Bradley T Hyman: Reviewing editor, *eLife*. The other authors declare that no competing interests exist.

### Funding

| Funder | Grant reference number | Author |
| --- | --- | --- |
| National Institute on Aging | NIA-K99AG068602 | Theodore J Zwang |
| Massachusetts General Hospital | Harrison Gardner, Jr Innovation Award | Theodore J Zwang |

The funders had no role in study design, data collection and interpretation, or the decision to submit the work for publication.

### Author contributions

Theodore J Zwang, Conceptualization, Resources, Data curation, Software, Formal analysis, Supervision, Funding acquisition, Validation, Investigation, Visualization, Methodology, Writing - original draft, Project administration, Writing - review and editing; Rachel E Bennett, Conceptualization, Resources, Data curation, Formal analysis, Supervision, Investigation, Methodology, Writing - review and editing; Maria Lysandrou, Benjamin Woost, Data curation, Formal analysis, Investigation, Writing - review and editing; Anqi Zhang, Conceptualization, Methodology, Writing - review and editing; Charles M Lieber, Conceptualization, Resources, Supervision, Methodology, Project administration, Writing - review and editing; Douglas S Richardson, Conceptualization, Resources, Data curation, Formal analysis, Methodology, Writing - original draft, Writing - review and editing; Bradley T Hyman, Conceptualization,

Resources, Data curation, Formal analysis, Supervision, Funding acquisition, Methodology, Project administration, Writing - review and editing

### Author ORCIDs
Theodore J Zwang (iD) http://orcid.org/0000-0002-6815-4482
Anqi Zhang (iD) http://orcid.org/0000-0001-6121-8095
Douglas S Richardson (iD) http://orcid.org/0000-0002-3189-2190
Bradley T Hyman (iD) http://orcid.org/0000-0002-7959-9401

### Ethics
Autopsy tissue from human brains were collected at Massachusetts General hospital, with informed consent of patients or their relatives and approval of local institutional review boards. Human tissue was provided by the Massachusetts Alzheimer's Disease Research Center (ADRC) with approval from the Mass General Brigham IRB (1999P009556).

This study was performed in strict accordance with the recommendations in the Guide for the Care and Use of Laboratory Animals of the National Institutes of Health. All of the mice were handled according to approved institutional animal care and use committee (IACUC protocol # 2019N000026 and 2020N000069).

### Decision letter and Author response
Decision letter https://doi.org/10.7554/eLife.84112.sa1
Author response https://doi.org/10.7554/eLife.84112.sa2

---

## Additional files

### Supplementary files
• Supplementary file 1. Table with additional information describing ADRC human tissue samples.

• Transparent reporting form

• MDAR checklist

• Source code 1. ImageJ macro 'SplitOirToTiff.ijm' to convert multichannel tiff to single channel prior to conversion to HDF5 for use in Ilastik.

• Source code 2. ImageJ macro 'ConvertTifToH5.ijm' to convert multipage tif to HDF5 for use in Ilastik.

• Source code 3. ImageJ macro 'ConvertH5toTiff.ijm' to convert Ilastik segmentation results from HDF5 to multipage tiff to for use in Matlab analysis code.

• Source code 4. Matlab code 'IlastikStainingIntensity.mlx' to input Ilastik or Imaris segmentation results in multipage tiff format and output staining signal, noise, percent, and signal-noise ratio for each imaging plane.

• Source code 5. Matlab code 'FindHalfMaxAndPlot.mlx' to run after 'IlastikStainingIntensity.mlx', which finds the imaging plane at the top of the sample, finds the half-max depth, unblinds data, then provides tables and plots that describe the results.

### Data availability
All data used in publication can be accessed at the BioImage Archive https://www. ebi.ac.uk/biostudies/studies/S-BIAD479. All Matlab code and ImageJ macros used to quantify data can be accessed at https://github.com/tjzwang/IHC, (copy archived at swh:1:rev:64acafda08ebdda1021edfe47e3d328b21218aa6).

The following dataset was generated:

| Author(s) | Year | Dataset title | Dataset URL | Database and Identifier |
|---|---|---|---|---|
| Zwang TJ, Bennett RE, Lysandrou M, Woost B, Zhang A, Lieber CM, Richardson DS, Hyman BT | 2022 | Tissue libraries enable rapid determination of conditions that preserve antibody labeling in cleared mouse and human tissue | https://www.ebi. ac.uk/biostudies/ bioimages/studies/ S-BIAD479 | ebi, S-BIAD479 |

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

com/tjzwang/IHC;visit=swh:1:snp:306b87154871c16f03dfdfa18f177cd2f6bfac0c;anchor=swh:1:rev:64acafda08ebdda1021edfe47e3d328b21218aa6

