## [Editor Report]

This valuable Tools and Resources paper presents a solid workflow for testing and comparing variations in tissue clearing, antigen retrieval, and antibody staining methods using thick slices of tissue. Though staining results vary sensitively with processing conditions, results from screening conditions in mouse brain tissue can be carried over to staining in human brain tissue. This solid story will be of broad interest to those carrying out immunohistochemistry experiments in human tissue samples.

---

## [Decision Letter]

**Decision letter after peer review:**

[Editors’ note: the authors submitted for reconsideration following the decision after peer review. What follows is the decision letter after the first round of review.]

Thank you for submitting the paper "Tissue libraries enable rapid determination of conditions that preserve antibody labeling in cleared mouse and human tissue" for consideration by *eLife*. Your article has been reviewed by 2 peer reviewers, and the evaluation has been overseen by a Reviewing Editor and a Senior Editor. The reviewers have opted to remain anonymous.

Comments to the Authors:

We are sorry to say that, after consultation with the reviewers, we have decided that this work will not be considered further for publication by *eLife* at this stage.

Specifically, the reviewers noted and detailed a lack of clarity in key points of the protocol, and a rather large variability in the staining outcomes. Additionally, the reviewers were concerned about a few key assumptions regarding the properties of the subset of antibodies tested, and premature generalization of the results from analysis of these antibodies.

Overall, the work has a lot of potential, and we encourage resubmission of the story when the reviewer comments can be adequately addressed.

The complete and detailed reviewer evaluations are below.

*Reviewer #1 (Recommendations for the authors):*

Immunohistochemistry (IHC) encompasses a complex set of methods with many possible variations on conditions for tissue fixation, clearing, antigen retrieval, blocking, and staining. Choices of these conditions that are optimized for one antibody/target pair often do not generalize to others, or to other tissue types. As high throughput microscopy (particularly lightsheet microscopy) and clearing methods allow for imaging thicker blocks of tissue, these problems are increasingly a limiting factor in experimental studies. A more systematic understanding of how these various aspects of IHC methods choices should be made is of great interest. Optimizing IHC conditions for a given protein target in a given tissue is currently somewhat of an art. A more systematic approach to carrying out this optimization is also potentially of great interest. This would be particularly helpful for human tissue, as it is too precious to routinely test many IHC method variants.

The authors set out to create a streamlined, standardized tissue processing and image analysis workflow for comparing different tissue preparations in terms of IHC performance. They create a library of mouse brain tissue specimens processed with several variants of each processing step including fixation, clearing, and antigen retrieval. They vary only one parameter at a time to facilitate direct comparison between conditions. They set up a data analysis pipeline that is extremely simple and easy to use but probably underpowered for comparing quantitatively between conditions because it keeps only a signal mask and discards signal intensity information. It is also not clear how the pipeline should be modified to accommodate targets with different types of staining pattern. The results are presented as a 1D heatmap that varies with depth in the specimen, rather than a simple line plot, making it difficult to compare quantitatively between conditions.

Each of 12 process variants is carried out for two antibodies on three replicate specimens to assess repeatability. The authors do not discuss this experiment in detail, but there is high variability between replicates. For example, in Figure S1, the 2 days cleared case has fairly high signal in two replicates and almost none in the middle one. The staining solutions were apparently also not well mixed, as the antibody signals on the two sides of each specimen are very different in all cases, and the amount of difference between the two sides is also highly varying between specimens. From Figure 1, it is also clear that the signal varies across each specimen within the imaging plane. In this comparison of 12 conditions, one result pops out: incubating the antibodies at room temperature (as opposed to the default of 4oC) improves penetration. This effect is dramatic for NeuN. It is probably also present for Glut1, although it is necessary to look at Figure S2, showing all three replicates, to see this clearly. Every other method variant appears to result in worse performance except that in the case of 2 day clearing with Glut1 staining, one replicate has a particularly high signal area (Figure S2).

They then test two antibodies against GFAP and show that varying processing conditions affects both antibodies similarly, though they do not include any quantification. In contrast, they find that two antibodies against tau do vary differently with processing conditions, but again without quantification. Next, they carry out the same screen in human brain tissue that had previously been done in mouse brain tissue in order to assess the suitability of mouse tissue as a proxy for IHC quality in human tissue. The trends of IHC quality as a function of method variant appear to be qualitatively preserved between mouse and human. They finally show that three antibodies (primaries directly conjugated with fluorophores) against Glut1, tau, and NeuN fully penetrate through a 1 mm slice of human brain tissue after incubating for 14 days.

The goal of putting IHC on a more quantitative footing is an admirable and timely one, but the approach presented here requires some more work to deliver on this substantial promise.

Tissue library: Tissue thickness and staining time choices should be justified more carefully. Of particular concern is the antibody staining time, as the optimal time depends strongly on the temperature of incubation. Staining should be quantified at several time points to determine the extent of saturation at the tissue surface. The assumption that antigens have a "relatively constant distribution with the depth of imaging" should also be supported with data. The same goes for lateral variation. Variation between replicates also would ideally be reduced. From the methods, it seems as though the antibody is simply added to the staining buffer with the tissue, which might indicate there is insufficient mixing before and during antibody incubation.

Image analysis: A more sophisticated but still highly user-friendly segmentation strategy such as the Ilastik software package might be a more reliable and versatile choice for analyzing image data. Furthermore, the signal intensity distribution within the masked signal area can be extracted (e.g. in Matlab) to quantitatively compare processing conditions, in addition to the area occupied by the mask as used here. The area outside the mask could potentially be treated as a background segment, whose intensity distribution can also be quantified. This way, quantitative measures of both signal intensity and background staining levels could be used for a more full comparison between method variants. The results should be viewed as a 1D line plot for more quantitative comparison between conditions. Finally, this lineplot could be used to measure derived performance parameters such as the distance into the tissue at which the staining intensity drops by half.

These improvements in quantitation are necessary to draw strong conclusions from any further experiments, but with them this work stands to make a strong contribution to the field.

*Reviewer #2 (Recommendations for the authors):*

This work presents a systematic approach to evaluation and optimization of antibody labeling in cleared human brain tissue. The authors outline the problems very clearly and present strategies to solve them. First, libraries of tissue prepared using different protocols are generated to easily test the performance of different antibodies and different staining conditions. Importantly, only one condition at a time is varied, which allows to evaluate the contribution of individual variables to the final staining quality. Second, a strategy for finding the optimal conditions for simultaneous staining with several different antibodies is outlined. And third and very important results is that mouse tissue can be used for optimization of antibody labeling of human tissue, thus allowing efficient use of human tissue which is a scarce and very valuable resource.

Strengths

Antibody labeling of tissues, and especially large volume cleared tissues involves a multitude of steps and reagents. Variations in any of the tissue preparation or immunolabeling steps can affect antibody performance and finding the optimal conditions can be a very tedious process. This paper provides clear examples of how much variability there can be even when using the same antibody on the same tissue prepared in different ways or stained under different conditions. It gives a clear solution of how this problem can be addressed by using libraries of tissues prepared by varying a single condition in each instance.

The presented approach results in a huge saving of time as it can be completed much faster (several days) compared to the complete immunolabeling protocol for large volume cleared tissue which can take several weeks.

The demonstration that mouse tissue can be used for optimization of staining conditions for human tissue is very important, allowing the best use of human tissue.

Weaknesses

Results from a small number of antibodies are extrapolated to apply to all antibodies in general. For example, the results from the comparison between 2 different GFAP antibodies, showing that the antibody conjugated to a fluorophore performs better that the antibody that is not conjugated, is used to generalize that primary antibodies directly conjugated to a fluorophore are preferrable to use compared to antibodies that are visualized using indirect immunolabeling with a secondary antibody. Even when considering the 2 GFAP antibodies this conclusion cannot be made, because it is not clear whether the difference in performance is due to the primary antibodies themselves (monoclonal vs. polyclonal) or to the method of their visualization (direct vs indirect).

The authors conclude that "IHC is limited more by denaturation of the epitope than the antibody itself". This conclusion is based on the experiments that show that varying the conditions of tissue preparation and staining can dramatically change antibody performance from no label to excellent label. However, the authors do not take into account that they are using antibodies that have already been shown to work reasonably well for immunolabeling.

Important methodological information is missing in some instances, for example how was the human tissue fixed or how are the tissue libraries stored.

Recommendations in order of appearance in the manuscript:

Introduction: It is very well written! It is important to specify here, as done further below in the paper (in Comparison of IHC conditions for different antibodies directed against a single protein) that the proposed strategies apply when "the antibodies are of high quality, have high specificity and limited off-target binding".

Results, Measuring IHC quality in cleared tissue: It is unclear how the threshold was determined. Another concern is that this "measure of IHC quality" is essentially a measure of what fraction of the tissue is labeled. Thus, it does not account for specific vs non-specific staining.

Conditions that influence antibody and epitope preservation: It is unclear whether the baseline protocol described here applies only to mouse tissue or to human tissue as well. The mention of perfusion leads me to believe this is specifically for mouse, but I couldn't find information on how the human tissue was fixed.

There is no mention here of whether the antibodies (GluT1 and NeuN) are directly conjugated to fluorophores. I assumed that they were, based on the info in the Methods section. But then in the last paragraph of Results the authors say that "Therefore, we directly conjugated three antibodies to three different fluorophores (NeuN-A568, Glut1- A488, and AT8-A647)." Which to me implies that they weren't conjugated in previous experiments.

Comparison of IHC conditions for different antibodies directed against a single protein: As explained in the Public review section, I disagree with the conclusion "it is recommended to use primary antibodies directly conjugated to fluorophores whenever possible." Unless the authors add more experiments that compare direct vs. indirect immmunolabeling with the same primary antibody (for several different antibodies), this conclusion is not justified.

"In order to examine how much of the observed variability in antibody labeling is due to targets being more or less accessible depending on cell type or subcellular location, multiple antibodies were also chosen that recognize tau protein." I don't understand what the comparison of the tau antibodies will tell us about the accessibility of targets depending on cell type or subcellular location.

Mouse brain tissue as a model system for optimizing IHC in cleared human tissue: Determining the optimal protocol for simultaneous staining of 3 antibodies is an important point. It will be very useful if the authors elaborate more on how this optimal protocol was determined. How was the combination of conditions that maximize staining quality established? Minor point: how was the 14 days duration determined?

Discussion: "Rather than the antibody, our data suggests that poor staining is most often due to denaturation or blockage of the epitope." I disagree with this conclusion, as explained above.

"Unfortunately, as we found with the dopamine receptor 2 and GFAP IHC, a protocol may not exist to allow for both targets to be effectively stained simultaneously." – I didn't see this presented in the Results section.

Materials and methods:

How were the human samples fixed? Were the slicing and all the following steps performed in an identical way? I assume they were, but this should be stated clearly in the methods. How were the tissue libraries stored? For how long can they be used after preparation?

Figures

Figures 1, 2 and others. How is the top of the side view determined? Is this the surface of the tissue and therefore conditions where the label in the side view starts lower that in other conditions means that the tissue surface was not labeled? And why would that be? If that is not the case, it will be best to align the side views and have the labeling start at the same level.

Figure S3. This is a very clear way of summarizing the differences in labeling. It would be very helpful to have this summary next to the actual immunostaining images in the main figures.

Figure S9 Representative microtome images taken from the human tissue prior to staining: Do you mean prior to clearing?

[Editors’ note: further revisions were suggested prior to acceptance, as described below.]

Thank you for resubmitting your work entitled "Tissue libraries enable rapid determination of conditions that preserve antibody labeling in cleared mouse and human tissue" for further consideration by *eLife*. Your revised article has been evaluated by Suzanne Pfeffer (Senior Editor) and a Reviewing Editor.

The manuscript has been improved but there are some remaining issues that need to be addressed, as outlined below:

Essential revisions:

Regarding the comparison between conjugated antibodies and primary plus secondary antibodies, it seems that it's possible to interpret the data in a different way, as outlined below by the reviewers, given that the comparison is not between the same primary antibodies with one conjugated and one not. Thus the observed differences can also be due to the primary antibodies themselves. Also, it is not clear whether a different secondary antibody would give a similar result.

On the question of antibody penetration, it is possible to interpret some of the results not as a problem of an impediment to physical penetration but of antibody depletion as the latter binds the tissue as it diffuses across.

*Reviewer #1 (Recommendations for the authors):*

The revised manuscript has addressed most of the reviewers' concerns and is significantly improved.Regarding the comparison between directly conjugated antibodies and use of secondary antibodies. I agree that directly conjugated antibodies save time and can be recommended for that purpose. I still have a problem with claims like: "As shown in Figure 3, the use of a primary antibody directly conjugated to a fluorophore greatly reduced background fluorescence over traditional primary/secondary antibody systems." The comparison is not between the same primary antibodies with one conjugated and one not; thus the observed differences can also be due to the primary antibodies themselves. Also, it is not at all clear whether a different secondary antibody would give a similar result. There is a great variety of secondary antibodies commercially available and, generally, the preadsorbed secondary antibodies that have undergone an extra purification step have higher specificity and give less background labeling; however the secondary antibodies used here were not of the "preadsorbed" kind.

The authors now include more details on the human tissue handling, which clarifies many questions that I had. A few remaining ones are:

Materials and methods: Human tissue samples: what was the postmortem interval before fixation? Even a rough estimate will be helpful.

Supplementary Table 1. Hemisphere column: "Frozen left". Does this mean that the human tissue was stored frozen? And then fixed? If it was indeed frozen before fixation, which was not mentioned elsewhere in the methods, that could explain why it looks like the antibodies penetrate deeper into the human tissue compared to mouse, (e.g. Figure 1 vs. Figure 5), but it is also an important difference to keep in mind.

*Reviewer #2 (Recommendations for the authors):*

The additions to the manuscript re-submission are greatly appreciated, in particular the updated analysis approach and 1d line profiles of signal versus tissue depth. There is one additional point that I believe is important and can be handled in the text. From the screen of antibody staining times (Figure S1), it appears that up through 24 hours of incubation the effect of increased incubation time is to increase the signal in a ~120 µm surface layer whose thickness does not change appreciably. After this, up to 48 hr (at least for RT incubation) the thickness of the surface layer changes strongly. This suggests that the limitation on signal for staining 24hr and under is depletion of antibodies on the way through the tissue. It should be pointed out that in the present work, the focus on depth to half-max staining does not separate the issues of (1) free antibody penetration speed through the tissue and (2) the extent of depletion along the way. Reducing antigenicity will improve the apparent penetration rate by reducing the effect of depletion due to epitope binding. This may be what is happening with the 50oC unmasking treatment, which increases penetration but greatly reduces the staining level (Figure 1 b,c, column 8). This limitation should be discussed explicitly.

---

## [Author Response]

[Editors’ note: the authors resubmitted a revised version of the paper for consideration. What follows is the authors’ response to the first round of review.]

Reviewer #1 (Recommendations for the authors):[…]Tissue library: Tissue thickness and staining time choices should be justified more carefully. Of particular concern is the antibody staining time, as the optimal time depends strongly on the temperature of incubation. Staining should be quantified at several time points to determine the extent of saturation at the tissue surface.

We have conducted additional experiments to compare room temperature and 4^o^C incubation of NeuN at 5 time points, plotted in figure S1. These results show both the increase in saturation at the tissue surface and within the tissue. For both incubation temperatures, the tissue appears to saturate at the surface within 24 hours but there are significant differences in the staining intensity with depth. Neither shows significant signal at a depth of 250 microns, suggesting that 500 micron thick tissue is adequate for comparing these conditions with <24 hour incubation times.

The assumption that antigens have a "relatively constant distribution with the depth of imaging" should also be supported with data. The same goes for lateral variation. Variation between replicates also would ideally be reduced.

We conducted additional experiments to compare NeuN and Glut1 staining across three different brain regions in longitudinal slices from each hemisphere of three different mice. We incubated these antibodies for 3 days to achieve saturation, then quantified the average signal, percent segmented area, average noise, and signal/noise ratio in each imaging plane. Our original metric of percent segmented area shows some variation with depth depending on the brain region, but we find that normalizing the signal intensity per segmented area allows for a quantifiable metric for staining quality that is independent on the change in distribution of antigen, so long as there is some antigen in each plane. These data show insignificantly different results across tissue samples and brain regions. Additionally, as we will describe below, Ilastik segmentation is more robust which decreased a significant amount of variation between replicates. We feel the variation between replicates is reasonable, as quantified by the Signal-Noise ratio standard error of mean values in the hippocampus (10.2±2.4%) anterior cortex (11.0±3.5%) and posterior cortex (10.3±1.8%), especially when considering these tissue samples came from different mice.

From the methods, it seems as though the antibody is simply added to the staining buffer with the tissue, which might indicate there is insufficient mixing before and during antibody incubation.

We have clarified in the methods section that the antibody was mixed in the staining buffer, and that the tissue was placed on a gentle shaker during antibody incubation for all conditions.

Image analysis: A more sophisticated but still highly user-friendly segmentation strategy such as the Ilastik software package might be a more reliable and versatile choice for analyzing image data.

We attempted to use Ilastik, as recommended, and found it to be an extremely robust solution for analyzing our data. Before, using Imaris, we did not have access to the batch processing module due to its cost. This meant that we had to go through each file and individually generate the segmentation mask for each channel, which was a significant amount of time for the hundreds of imaging files used in this work. Ilastik allowed for us to train the segmentation on a subset of our data, then apply the segmentation to all of the similar images. Additionally, this did not require the normalization step we had applied to our data in Imaris, further improved the segmentation of poorly stained samples, and allowed for the isolation of nonspecific staining that may have confounded analysis. We are extremely grateful for this suggestion.

Furthermore, the signal intensity distribution within the masked signal area can be extracted (e.g. in Matlab) to quantitatively compare processing conditions, in addition to the area occupied by the mask as used here. The area outside the mask could potentially be treated as a background segment, whose intensity distribution can also be quantified. This way, quantitative measures of both signal intensity and background staining levels could be used for a more full comparison between method variants. The results should be viewed as a 1D line plot for more quantitative comparison between conditions. Finally, this lineplot could be used to measure derived performance parameters such as the distance into the tissue at which the staining intensity drops by half.

We appreciate these very constructive and insightful suggestions. As suggested, we performed this analysis on four of our sets of data (Mouse NeuN, Mouse Glut1, Human NeuN, and Human Glut1) consisting of 142 image volumes as a proof-of-principle. These quantifications provide some additional insight into the difference between conditions. Most interesting is the significant difference in the percent staining area in each imaging plane was more changed than the signal intensity across conditions, which provides support for our claim that IHC is significantly influenced by denaturation or masking of the epitope. The antibody is having the same overall effectiveness for signal intensity per unit area, but is binding to less. Additionally, the performance parameter suggested by the reviewer (which we call half-max depth in the manuscript) showed reproducible differences across samples and translated well between mouse and human tissue. We included this plot in Figure 3 and Figure 7.

Reviewer #2 (Recommendations for the authors):[…]WeaknessesResults from a small number of antibodies are extrapolated to apply to all antibodies in general. For example, the results from the comparison between 2 different GFAP antibodies, showing that the antibody conjugated to a fluorophore performs better that the antibody that is not conjugated, is used to generalize that primary antibodies directly conjugated to a fluorophore are preferrable to use compared to antibodies that are visualized using indirect immunolabeling with a secondary antibody. Even when considering the 2 GFAP antibodies this conclusion cannot be made, because it is not clear whether the difference in performance is due to the primary antibodies themselves (monoclonal vs. polyclonal) or to the method of their visualization (direct vs indirect).

Although we have used relatively few antibodies compared to the many thousands used in research, we rigorously tested each of the antibodies in this study. We have collected a large amount of data for each antibody, which includes at least three replicates from independent brains for 12 different conditions. Additionally, each replicate includes approximately 100 imaging planes with 2-4 channels, depending on the stains that were tested. We feel confident that these data strongly support our main claims that small changes in conditions lead to significant variation in IHC quality, which we see among the antibodies we tested. Additionally, the variation in IHC quality for a dozen different conditions is consistent between mice and human tissue that are prepared similarly. These results are what allowed for us to find conditions that produce the beautiful images in figure 8 that show simultaneous staining of multiple antibodies in human tissue. That said, we did not rigorously test the use of conjugated primary antibodies compared to use with secondary antibodies with multiple antibodies, so we have removed those claims which we feel are a minor point in the original manuscript.

The authors conclude that "IHC is limited more by denaturation of the epitope than the antibody itself". This conclusion is based on the experiments that show that varying the conditions of tissue preparation and staining can dramatically change antibody performance from no label to excellent label. However, the authors do not take into account that they are using antibodies that have already been shown to work reasonably well for immunolabeling.

We believe that the new analyses suggested by reviewer 1 help support our claim that IHC may be limited more by the denaturation of the epitope than the antibody itself. We agree that we are using antibodies that have already been shown to work reasonably well for immunolabeling, and have included mention of this in the second paragraph of the Results section.

Important methodological information is missing in some instances, for example how was the human tissue fixed or how are the tissue libraries stored.

We have included more details of our methodology in the *Human tissue samples* and *Immunohistochemistry* sections, as well as added a section on the *Storage of tissue library*

Recommendations in order of appearance in the manuscript:Introduction: It is very well written! It is important to specify here, as done further below in the paper (in Comparison of IHC conditions for different antibodies directed against a single protein) that the proposed strategies apply when "the antibodies are of high quality, have high specificity and limited off-target binding".Results, Measuring IHC quality in cleared tissue: It is unclear how the threshold was determined. Another concern is that this "measure of IHC quality" is essentially a measure of what fraction of the tissue is labeled. Thus, it does not account for specific vs non-specific staining.

We believe that the new methodology using ilastik addresses reviewer 2’s concerns as well. Using this new methodology there is no need for setting a threshold. It does require the users to properly train the segmentation software, however the results are updated live during training which allows for users to quickly assess its quality. Furthermore, you can train ilastik to separate non-specific staining from the signal and background which allows for quantification of non-specific staining.

Conditions that influence antibody and epitope preservation: It is unclear whether the baseline protocol described here applies only to mouse tissue or to human tissue as well. The mention of perfusion leads me to believe this is specifically for mouse, but I couldn't find information on how the human tissue was fixed.

We have included more details of this methodology in the *Human tissue samples* and *Immunohistochemistry* sections.

There is no mention here of whether the antibodies (GluT1 and NeuN) are directly conjugated to fluorophores. I assumed that they were, based on the info in the Methods section. But then in the last paragraph of Results the authors say that "Therefore, we directly conjugated three antibodies to three different fluorophores (NeuN-A568, Glut1- A488, and AT8-A647)." Which to me implies that they weren't conjugated in previous experiments.

We have included more details of this methodology in *Immunohistochemistry* section, as well as in the figure 3 and 7 captions.

Comparison of IHC conditions for different antibodies directed against a single protein: As explained in the Public review section, I disagree with the conclusion "it is recommended to use primary antibodies directly conjugated to fluorophores whenever possible." Unless the authors add more experiments that compare direct vs. indirect immmunolabeling with the same primary antibody (for several different antibodies), this conclusion is not justified.

As recommended by reviewer 2, we have withdrawn the conclusion that our data shows significant advantages of primary over secondary antibodies for reducing nonspecific staining. This would take a significant amount of effort for a relatively minor point, which we may be able to address in the future, however we are currently exclusively using conjugated antibodies in all our experiments. We have still included a recommendation to use conjugated antibodies where possible because it reduces the number of steps and total time for IHC, sometimes on the order of weeks depending on tissue thickness.

"In order to examine how much of the observed variability in antibody labeling is due to targets being more or less accessible depending on cell type or subcellular location, multiple antibodies were also chosen that recognize tau protein." I don't understand what the comparison of the tau antibodies will tell us about the accessibility of targets depending on cell type or subcellular location.

We have removed our statement on subcellular location, but hope that we can address this in future work.

Mouse brain tissue as a model system for optimizing IHC in cleared human tissue: Determining the optimal protocol for simultaneous staining of 3 antibodies is an important point. It will be very useful if the authors elaborate more on how this optimal protocol was determined. How was the combination of conditions that maximize staining quality established?

We have clarified our process for determining the optimal protocol from the collected tissue library data in the main text in multiple places, including the following text under Mouse Brain tissue as a model system for optimizing IHC in cleared human tissue in the Results section: “The optimal protocol that allows NeuN, Glut1, and AT8-tau staining was then determined by finding the conditions that allow for the simultaneous IHC of all three antibodies in a single tissue section, and then finding the combination of conditions among those that maximizing the staining quality. It is worth noting that this was not necessarily the best protocol for each individual antibody. It is more important to have adequate staining of each epitope than to have fantastic staining of one at the cost of another. As described above, we determined the optimal protocol for multiplexed labeling by combining the protocol changes that quantitatively improve the staining quality, defined by half-max depth, for each antibody above the baseline protocol and do not significantly degrade performance of other antibodies. For NeuN, Glut1, and AT8 antibodies this protocol involves clearing for 2 days, using 50^o^C to unmask epitopes, and incubation of antibodies at room temperature.”

Discussion: "Rather than the antibody, our data suggests that poor staining is most often due to denaturation or blockage of the epitope." I disagree with this conclusion, as explained above.

We believe that the new analyses suggested by reviewer 1 help support our claim that IHC may be limited more by the denaturation of the epitope than the antibody itself. We agree that we are using antibodies that have already been shown to work reasonably well for immunolabeling, and have included mention of this in the second paragraph of the Results section.

"Unfortunately, as we found with the dopamine receptor 2 and GFAP IHC, a protocol may not exist to allow for both targets to be effectively stained simultaneously." – I didn't see this presented in the Results section.

It is described in the section Conditions that influence antibody and epitope preservation in the second paragraph, specifically: “Inclusion of PFA in the hydrogel resulted in decreased IHC quality for both Glut1 and NeuN, however this significantly improved IHC of dopamine receptor 2 (figure 4-supplement 1).”

Materials and methods:How were the human samples fixed? Were the slicing and all the following steps performed in an identical way? I assume they were, but this should be stated clearly in the methods. How were the tissue libraries stored? For how long can they be used after preparation?

We have added a new section “Storage of tissue library” with the following text: The tissue library was then used immediately or stored at 4^o^C in PBS with 0.02% (w/v) sodium azide. Mouse and human tissue was processed following all of the steps above and then used up to 3 months later. Additional mouse and human tissue was stored following delipidation, then successfully used up to 6 months later by resuming from the pre-immunohistochemistry tissue processing state. IHC conditions for a given antibody were always compared using batches of tissue that were stored for the same amount of time.

FiguresFigures 1, 2 and others. How is the top of the side view determined? Is this the surface of the tissue and therefore conditions where the label in the side view starts lower that in other conditions means that the tissue surface was not labeled? And why would that be? If that is not the case, it will be best to align the side views and have the labeling start at the same level.

We have aligned the side views as recommended by reviewer 2. In these images, sometimes the imagine plane started slightly above the top plane of the tissue. For the purposes of quantification and presentation, our matlab code now starts the plot from the first plane with 80% of the max staining intensity within a sample. This is clarified in the Materials and methods where we describe the code for analysis. The top view is chosen arbitrarily as a section that faithfully represents the quality of the rest of the image.

Figure S3. This is a very clear way of summarizing the differences in labeling. It would be very helpful to have this summary next to the actual immunostaining images in the main figures.

We have added the half-max intensity values to figure 3 near the actual immunostaining images to summarize the differences in labeling.

Figure S9 Representative microtome images taken from the human tissue prior to staining: Do you mean prior to clearing?

In figure S9 (now figure 6-supplement 2) we do mean that the images were taken prior to clearing, and have changed the text to clarify that point.

[Editors’ note: what follows is the authors’ response to the second round of review.]

Essential revisions:Reviewer #1 (Recommendations for the authors):The revised manuscript has addressed most of the reviewers' concerns and is significantly improved.Regarding the comparison between directly conjugated antibodies and use of secondary antibodies. I agree that directly conjugated antibodies save time and can be recommended for that purpose. I still have a problem with claims like: "As shown in Figure 3, the use of a primary antibody directly conjugated to a fluorophore greatly reduced background fluorescence over traditional primary/secondary antibody systems." The comparison is not between the same primary antibodies with one conjugated and one not; thus the observed differences can also be due to the primary antibodies themselves. Also, it is not at all clear whether a different secondary antibody would give a similar result. There is a great variety of secondary antibodies commercially available and, generally, the preadsorbed secondary antibodies that have undergone an extra purification step have higher specificity and give less background labeling; however the secondary antibodies used here were not of the "preadsorbed" kind.

As recommended by the reviewer, we have removed any claims that suggest an improvement in the staining from using conjugated antibodies. Specifically, we have replaced the following sentences from the previous draft: 

“Importantly, primary GFAP antibodies directly conjugated to a fluorophore showed significantly better penetration, signal-to-noise ratio, and less off-target staining for both GFAP antibodies under all conditions where IHC successfully labeled its target (Figure 3) and both showed similar conditions where no labeling occurred (Figure s10).”

To these new sentences in the revised draft:

“We found that both GFAP antibodies successfully labeled their target under similar conditions, (Figure 3) and both showed similar conditions where no labeling occurred (Figure s10). Some differences in labeling quality are apparent between antibodies, however it is not clear whether the differences in labeling quality are due to the conjugation of the fluorophore or other differences between the monoclonal and polyclonal antibodies.”

We also removed the following sentence: “As shown in Figure 3, the use of a primary antibody directly conjugated to a fluorophore greatly reduced background fluorescence over traditional primary/secondary antibody systems.”

Last paragraph of Results before Discussion: "Therefore, we directly conjugated three antibodies to three different fluorophores (NeuN-A568, Glut1-A488, and AT8-A647)." This seems to be incorrect as these are the commercially available antibodies that are sold already directly conjugated.

Thank you for catching this mistake. We changed the sentence to the following: “We concurrently incubated the human tissue with three antibodies conjugated to three different fluorophores (NeuNA568, Glut1-A488, and AT8-A647).”

Materials and methods: Human tissue samples: what was the postmortem interval before fixation? Even a rough estimate will be helpful.We have amended supplementary table 1 to include the postmortem interval before fixation.Supplementary Table 1. Hemisphere column: "Frozen left". Does this mean that the human tissue was stored frozen? And then fixed? If it was indeed frozen before fixation, which was not mentioned elsewhere in the methods, that could explain why it looks like the antibodies penetrate deeper into the human tissue compared to mouse, (e.g. Figure 1 vs. Figure 5), but it is also an important difference to keep in mind.

To help clarify that some of the human tissue was stored frozen then thawed prior to fixation we have added the following sentence to the Human tissue samples section under Materials and methods: “Human tissue was received unfrozen immediately after autopsy and fixed (sample 2620) or frozen immediately after autopsy then thawed prior to fixation (samples 1581 and 1762).”

Reviewer #2 (Recommendations for the authors):The additions to the manuscript re-submission are greatly appreciated, in particular the updated analysis approach and 1d line profiles of signal versus tissue depth. There is one additional point that I believe is important and can be handled in the text. From the screen of antibody staining times (Figure S1), it appears that up through 24 hours of incubation the effect of increased incubation time is to increase the signal in a ~120 µm surface layer whose thickness does not change appreciably. After this, up to 48 hr (at least for RT incubation) the thickness of the surface layer changes strongly. This suggests that the limitation on signal for staining 24hr and under is depletion of antibodies on the way through the tissue. It should be pointed out that in the present work, the focus on depth to half-max staining does not separate the issues of (1) free antibody penetration speed through the tissue and (2) the extent of depletion along the way. Reducing antigenicity will improve the apparent penetration rate by reducing the effect of depletion due to epitope binding. This may be what is happening with the 50oC unmasking treatment, which increases penetration but greatly reduces the staining level (Figure 1 b,c, column 8). This limitation should be discussed explicitly.

We would like to thank Reviewer #2 for the insight into our data, as well as their assessment of its limitations. We have added the following text to explicitly reflect their insight: 

We added the following text under the “Strategy to rapidly assess the optimization of protocols” section in paragraph 2 “It should be noted that these data suggest the signal when staining 24 hours and under may be limited by the depletion of antibodies on their way through the tissue, which should be considered when making comparisons of staining quality.”

We added the following text under the “Measuring IHC quality in cleared tissue” It should be noted that simply relying on depth to half-max staining does not separate the issues of free antibody penetration speed through the tissue nor the extent of antibody depletion along the way, and it is possible that conditions that reduce the antigenicity may significantly enhance the apparent penetration rate.